# THE NEURAL DATA ROUTER: ADAPTIVE CONTROL FLOW IN TRANSFORMERS IMPROVES SYSTEMATIC GENERALIZATION

**Róbert Csordás**[1]  **Kazuki Irie**[1]  **Jürgen Schmidhuber**[1,2]
[1]The Swiss AI Lab, IDSIA, University of Lugano (USI) & SUPSI, Lugano, Switzerland
[2]King Abdullah University of Science and Technology (KAUST), Thuwal, Saudi Arabia
{robert, kazuki, juergen}@idsia.ch

## ABSTRACT

Despite progress across a broad range of applications, Transformers have limited success in systematic generalization. The situation is especially frustrating in the case of algorithmic tasks, where they often fail to find intuitive solutions that route relevant information to the right node/operation at the right time in the grid represented by Transformer columns. To facilitate the learning of useful control flow, we propose two modifications to the Transformer architecture, copy gate and geometric attention. Our novel Neural Data Router (NDR) achieves 100% length generalization accuracy on the classic compositional table lookup task, as well as near-perfect accuracy on the simple arithmetic task and a new variant of ListOps testing for generalization across computational depths. NDR's attention and gating patterns tend to be interpretable as an intuitive form of *neural routing*. Our code is public.[1]

## 1 INTRODUCTION

Neural networks (NNs) may easily learn certain training sets, but typically they do not generalize on systematically different test sets. Examples of systematic generalization (Fodor et al., 1988)

include generalization to sequences longer than those seen during training—productivity, and algorithmic combinations of previously learned rules—systematicity. Despite recent efforts (Bahdanau et al., 2019; Korrel et al., 2019; Lake, 2019; Li et al., 2019; Russin et al., 2019; Csordás et al., 2021), systematic generalization generally remains unsolved (Fodor & McLaughlin, 1990; Lake & Baroni, 2018; Liska et al., 2018; Greff et al., 2020; Hupkes et al., 2020). On some datasets, the best performing models are neuro-symbolic hybrids (Chen et al., 2020; Liu et al., 2020) using task-specific symbolic functions. However, their applicability to other datasets remains limited (Furrer et al., 2020; Shaw et al., 2020). A big question is: which type of architectural inductive bias encourages the training process to select "good" solutions which generalize systematically?

The popular Transformers (Vaswani et al., 2017) also often fail to generalize on algorithmic tasks (e.g. Liska et al. (2018); Dubois et al. (2020); Chaabouni et al. (2021); Csordás et al. (2021); Ontañón et al. (2021)), even on tasks with intuitive solutions that can be simply expressed in terms of Transformer attention patterns. Given an input sequence of length $N$ and a Transformer encoder of depth $T$, solving an algorithmic task is often all about routing the relevant information to the right node/operation at the right time in the $T$-by-$N$ grid represented by Transformer columns (illustrated in Figure 1/Left). Effectively the task is to learn to draw an *adaptive control flow* on the canvas of Transformer columns. In fact, recent work by Weiss et al. (2021) introduced a programming language called RASP, which is specifically designed to express solutions to sequence processing problems, and which has a direct equivalent to the operations in Transformer encoders. However, it is shown that Transformers learn solutions expressed in RASP only through intermediate supervision of attention patterns, and sometimes, even such supervision fails. Generally speaking, Transformers fail to find easily interpretable and/or symbolic solutions to algorithmic tasks. We conversely hypothesize that attention-based NNs that are able to find intuitive solutions (achieving interpretable attention patterns) could improve systematic generalization.

---

[1]https://github.com/robertcsordas/ndr

Here we point out that regular Transformers lack some basic ingredients for learning such "intuitive" solutions to algorithmic problems. As a remedy, we propose simple architectural modifications to help them learn data routing. As a first step towards validating our model, we focus on the popular length generalization task of compositional table lookup (CTL; Liska et al. (2018); Hupkes et al. (2019); Dubois et al. (2020)), as well as two more complex tasks: a simple arithmetic task and a variant of ListOps (Nangia & Bowman, 2018) designed to test the compositional generalization ability of NNs. Our novel Neural Data Router (NDR) achieves 100% generalization accuracy (never reported before; Dubois et al. (2020)) on the CTL task, and obtains nearly perfect accuracy on both the proposed simple arithmetic and ListOps tasks. We show that the attention and gating patterns of NDR tend to be interpretable as plausible control flows.

## 2 Improving Transformers for Learning Adaptive Control Flow

We argue that the following components are needed to build Transformers capable of learning adaptive control flow. **First**, composing known operations in an arbitrary order requires that all operations are available at every computational step. This can be easily achieved by sharing the weights of the layers, as is done in Universal Transformers (Dehghani et al., 2019). **Second**, the network should be sufficiently deep, at least as deep as the deepest data dependency in the computational graph built from elementary operations (e.g., in the case of a parse tree, this is the depth of the tree). Otherwise, multiple operations must be fused into a single layer and hinder natural and elegant compositions. **Third**, inputs in some columns should be kept unchanged until it is their turn to be processed. The regular Transformer lacks a mechanism for skipping the whole transformation step by simply copying the input to the next step/layer. We propose a special gating function, *copy gate*, to implement such a mechanism (Sec. 2.1). **Finally**, many algorithmic tasks require combining several local computations in the right order. This typically implies that attention should not focus on all possible matches at a given time but only on the closest match. We propose and investigate a new type of attention with a corresponding inductive bias called *geometric attention* (Sec. 2.2). Using both the geometric attention and copy gate, our model implements a "neural data routing mechanism", which can adaptively serialize the input problem. We refer to the resulting new Transformer as Neural Data Router (NDR). In the experimental section (Sec. 3), we evaluate this model on three algorithmic tasks requiring length generalization and demonstrate its effectiveness.

### 2.1 Copy Gate: Learning to Skip Operations (Vertical Flow)

Each layer of the regular Transformer consists of one self-attention and one feedforward block. The input to each of these blocks is directly connected to the corresponding output via a residual connection (Srivastava et al., 2015; He et al., 2016). However, such a connection does not allow for skipping the transformation of the entire layer and simply passing the unchanged input to the next layer. Here we propose to add an explicit gate, which we call *copy gate*, to facilitate such a behavior.

We consider a $T$-layer Transformer encoder and an input sequence of length $N$. Since each layer corresponds to one *computational step*, we often refer to a layer as a step $t$. We denote the Transformer state of column $i$ in layer $t$ as $\boldsymbol{h}^{(i,t)} = \mathsf{H}_{t,i} \in \mathbb{R}^d$ where $d$ is the state size, and $\mathsf{H}_t \in \mathbb{R}^{N \times d}$ denotes the states of all $N$ columns in layer $t$. In the copy gate-augmented Transformer (Figure 5 in the appendix), each column $i$ in layer $(t+1)$ processes the input $\mathsf{H}_t$ similarly to regular Transformers:

$$\boldsymbol{a}^{(i,t+1)} = \mathrm{LayerNorm}(\mathrm{MultiHeadAttention}(\boldsymbol{h}^{(i,t)}, \mathsf{H}_t, \mathsf{H}_t) + \boldsymbol{h}^{(i,t)}) \tag{1}$$

$$\boldsymbol{u}^{(i,t+1)} = \mathrm{LayerNorm}(\mathrm{FFN}^{\mathrm{data}}(\boldsymbol{a}^{(i,t+1)})) \tag{2}$$

using the standard multi-head attention operation (Vaswani et al., 2017) $\mathrm{MultiHeadAttention}$ with a query obtained from $\boldsymbol{h}^{(i,t)}$ and keys/values from $\mathsf{H}_t$, but the output is gated (using $\boldsymbol{g}^{(i,t+1)} \in \mathbb{R}^d$) as:

$$\boldsymbol{g}^{(i,t+1)} = \sigma(\mathrm{FFN}^{\mathrm{gate}}(\boldsymbol{a}^{(i,t+1)})) \tag{3}$$

$$\boldsymbol{h}^{(i,t+1)} = \boldsymbol{g}^{(i,t+1)} \odot \boldsymbol{u}^{(i,t+1)} + (1 - \boldsymbol{g}^{(i,t+1)}) \odot \boldsymbol{h}^{(i,t)} \tag{4}$$

We use the basic two-layer feedforward block (Vaswani et al., 2017) for both $\mathrm{FFN}^{\mathrm{data}}$ and $\mathrm{FFN}^{\mathrm{gate}}$ which transforms input $\boldsymbol{x} \in \mathbb{R}^d$ to:

$$\mathrm{FFN}(\boldsymbol{x}) = \boldsymbol{W}_2 \max(\boldsymbol{W}_1 \boldsymbol{x} + \boldsymbol{b}_1, 0) + \boldsymbol{b}_2 \tag{5}$$

but with separate parameters and different dimensionalities: for $\text{FFN}^{\text{data}}$ $\boldsymbol{W}_1^{\text{data}} \in \mathbb{R}^{d_{\text{FF}} \times d}$, $\boldsymbol{W}_2^{\text{data}} \in \mathbb{R}^{d \times d_{\text{FF}}}$, while for $\text{FFN}^{\text{gate}}$ $\boldsymbol{W}_1^{\text{gate}}, \boldsymbol{W}_2^{\text{gate}} \in \mathbb{R}^{d \times d}$, with biases $\boldsymbol{b}_1^{\text{data}} \in \mathbb{R}^{d_{\text{FF}}}$ and $\boldsymbol{b}_2^{\text{data}}, \boldsymbol{b}_1^{\text{gate}}, \boldsymbol{b}_2^{\text{gate}} \in \mathbb{R}^d$.

When the gate is closed i.e. $\boldsymbol{g}^{(i,t+1)} = 0$ in Eq. 4, the entire transformation is skipped and the input is copied over to the next layer $\boldsymbol{h}^{(i,t+1)} = \boldsymbol{h}^{(i,t)}$. Crucially, we parameterize the gate (Eq. 3) as a function of the output of the self-attention (Eq. 1), such that the decision to copy or transform the input for each column depends on the states of all columns. This is a crucial difference compared to previously proposed gatings in Transformers, which are solely motivated by training stability (Parisotto et al., 2020) or by a common practice from convolution-based models (Chaabouni et al., 2021). None of the previous approaches can implement the behavior of our copy gate (see Sec. 6 on related work).

The bias of the gate $\boldsymbol{b}_2^{\text{gate}}$ is initialized to $-3$ (Hochreiter & Schmidhuber, 1997). This ensures that no update happens initially to create a better gradient flow between layers. It also encourages the model to skip layers unless they have an important contribution in the corresponding step.

## 2.2 GEOMETRIC ATTENTION: LEARNING TO ATTEND TO THE CLOSEST MATCH (HORIZONTAL FLOW)

We propose *geometric attention* designed to attend to the closest matching element. Like in regular self-attention, given an input sequence $[\boldsymbol{x}^{(1)}, \boldsymbol{x}^{(2)}, ..., \boldsymbol{x}^{(N)}]$ with $\boldsymbol{x}^{(i)} \in \mathbb{R}^{d_{\text{in}}}$, each input is projected to key $\boldsymbol{k}^{(i)} \in \mathbb{R}^{d_{\text{key}}}$, value $\boldsymbol{v}^{(i)} \in \mathbb{R}^{d_{\text{value}}}$, query $\boldsymbol{q}^{(i)} \in \mathbb{R}^{d_{\text{key}}}$ vectors, and the dot product is computed for each key/query combination. In our geometric attention, the dot product is followed by a sigmoid function to obtain a score between 0 and 1:

$$\boldsymbol{P}_{i,j} = \sigma(\boldsymbol{k}^{(j)\top}\boldsymbol{q}^{(i)}) \tag{6}$$

which will be treated as a probability of the key at (source) position $j$ matching the query at (target) position $i$. These probabilities are finally converted to the attention scores $\boldsymbol{A}_{i,j}$ as follows:

$$\boldsymbol{A}_{i,j} = \boldsymbol{P}_{i,j} \prod_{k \in \mathbb{S}_{i,j}} (1 - \boldsymbol{P}_{i,k}) \tag{7}$$

where $\mathbb{S}_{i,j}$ denotes the set of all (source) indices which are closer to $i$ than $j$ is to $i$, and when two indices have the same distance to $i$, we consider the one which is to the right of $i$ (i.e., greater than $i$) to be closer, i.e.,

$$\mathbb{S}_{i,j} = \begin{cases} k \in \{1, ..., N\} \setminus \{i, j\} : |i - k| < |i - j|, & \text{if } i < j \\ k \in \{1, ..., N\} \setminus \{i, j\} : |i - k| \le |i - j|, & \text{if } j < i \end{cases} \tag{8}$$

In addition, we explicitly zero out the diagonal by setting $\boldsymbol{A}_{i,i} = 0$ for all $i = 1, ..., N$. The ordering of source indices is illustrated in Figure 1/Right. The resulting scores $\boldsymbol{A}_{i,j}$ are the attention scores used to compute the weighted averages of the value vectors.

By using the terms $(1 - \boldsymbol{P}_{i,k})$ in Eq. 7, when there is a match, it downscales any other more distant matches. Two recent works (Brooks et al., 2021; Banino et al., 2021) use such a parameterized geometric distribution in the form of Eq. 7 (see Sec. 6 on related work).

The resulting attention function has a complexity of $O(N^2)$, similar to the regular self-attention used in Transformers (Vaswani et al., 2017). Eq. 7 can be implemented in a numerically stable way in log space. The products can then be calculated using cumulative sums, subtracting the elements for the correct indices in each position.

**Directional encoding.** In practice, we augment Eq. 6 with an additional *directional encoding*. In fact, the only positional information available in the geometric attention presented above is the ordering used to define the product in Eqs. 7-8. In practice, we found it crucial to augment the score computation of Eq. 6 with additional *directional information*, encoded as a scalar $\boldsymbol{D}_{i,j} \in \mathbb{R}$ for each target/source position pair $(i,j)$:

$$\boldsymbol{D}_{i,j} = \begin{cases} \boldsymbol{W}_{\text{LR}}\boldsymbol{h}^{(i)} + b_{\text{LR}}, & \text{if } i \le j \\ \boldsymbol{W}_{\text{RL}}\boldsymbol{h}^{(i)} + b_{\text{RL}}, & \text{if } i > j \end{cases} \tag{9}$$

where $\boldsymbol{h}^{(i)} \in \mathbb{R}^d$ denotes the input/state at position $i$ and $\boldsymbol{W}_{\text{LR}}, \boldsymbol{W}_{\text{RL}} \in \mathbb{R}^{1 \times d}$, $b_{\text{LR}}, b_{\text{RL}} \in \mathbb{R}$ are trainable parameters. This directional information is integrated into the score computation of Eq. 6 as

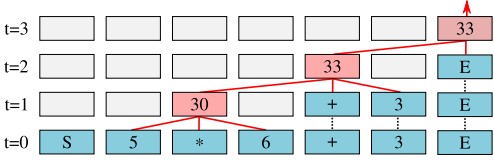 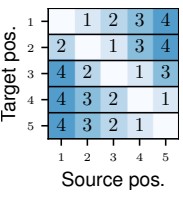

Figure 1: Left: an ideal sequence of computations in a Transformer for an arithmetic expression. Right: ordering (numbers in the grid) of source positions used in geometric attention (Eq. 8; $N = 5$).

follows (akin to how Dai et al. (2019) introduce the relative positional encoding (Schmidhuber, 1992) as an extra term in the computation of attention scores):

$$\boldsymbol{P}_{i,j} = \sigma\big(\alpha\big(\boldsymbol{W}_q \boldsymbol{h}^{(i)} + \boldsymbol{b}_q\big)^\top \boldsymbol{W}_{k,E} \boldsymbol{h}^{(j)} + \beta \boldsymbol{D}_{i,j} + \gamma\big) \tag{10}$$

where the matrix $\boldsymbol{W}_q \in \mathbb{R}^{d_{\text{head}} \times d}$ maps the states to queries, $\boldsymbol{b}_q \in \mathbb{R}^{d_{\text{head}}}$ is a bias for queries, $\boldsymbol{W}_{k,E} \in \mathbb{R}^{d_{\text{head}} \times d}$ maps states to keys (we note that $d_{\text{head}}$ is typically the size of the key, query and value vectors for each head, $d_{\text{head}} = \frac{d}{n_{\text{heads}}}$), and $\alpha, \beta, \gamma \in \mathbb{R}$ are learned scaling coefficients and bias, initialized to $\alpha = \frac{1}{\sqrt{d_{\text{head}}}}, \beta = 1, \gamma = 0$. Using this additional directional information, each query (position $i$) can potentially learn to restrict its attention to either the left or right side.

## 3 EXPERIMENTS

We evaluate the proposed methods on three tasks: the compositional table lookup (Liska et al., 2018; Hupkes et al., 2019), a custom variant of ListOps (Nangia & Bowman, 2018), and a simple arithmetic task which we propose. In all cases, the task is designed to test the compositional generalization ability of NNs: the model has to learn to apply operations seen during training in a longer/deeper compositional way (productivity). Further experimental details for each task can be found in Appendix C.

### 3.1 COMPOSITIONAL TABLE LOOKUP

**Task.** The compositional table lookup task (Liska et al., 2018; Hupkes et al., 2019; Dubois et al., 2020) is constructed based on a set of symbols and unary functions defined over these symbols. Each example in the task is defined by one input symbol and a list of functions to be applied sequentially, i.e., the first function is applied to the input symbol and the resulting output becomes the input to the second function, and so forth. There are eight possible symbols. Each symbol is traditionally represented by a 3-bit bitstring (Liska et al., 2018). However, in practice, they are simply processed as one token (Dubois et al., 2020). The functions are bijective and randomly generated. Each function is represented by a letter. An example input is '101 d a b', which corresponds to the expression $b(a(d(101)))$; the model has to predict the correct output symbol. We note that there exists a sequence-to-sequence variant of this task (Dubois et al., 2020) where the model has to predict all intermediate steps (thus trained with intermediate supervision). We directly predict the final output. An ideal model should be able to solve this task independently of the presentation order, that is, it should not matter whether the task is encoded as '101 d a b' or 'b a d 101'. We thus study both forward (former) and backward (latter) variants of the task. To evaluate systematic generalization, the train/valid/test sets reflect different numbers of compositions: samples with 1-5/6-8/9-10 operations, respectively. To best of our knowledge, no previous work has reported perfect accuracy on this task through an NN. We refer the readers to Sec. 6 for further details on the previous work.

**Results.** We consider five different baselines: an LSTM (Hochreiter & Schmidhuber, 1997), bidirectional LSTM (Schuster & Paliwal, 1997), DNC (Graves et al., 2016; Csordás & Schmidhuber, 2019), Universal Transformers (Vaswani et al., 2017; Dehghani et al., 2019), and its relative position variants (Csordás et al., 2021). For Transformers, the prediction is based on the last column in the final layer (we conduct an ablation study on this choice in Appendix A). The hyper-parameters used for each model can be found in Table 7 in the appendix. We also provide an ablation study on the number of layers needed for generalization in Appendix A, which supports our claim on the necessity

for a "sufficiently" deep architecture. The main results on this task are shown in Table 1. The LSTM and DNC perform well in the forward variant, achieving perfect generalization for longer sequences, but fail on the backward variant. This is not surprising since in the forward case, input symbols are presented in the "right" processing order to the LSTM. As expected, the bidirectional LSTM performs well in both presentation orders, since one of its processing directions is always aligned with the order of computation. However, for an arbitrary task, the order of processing is not given. For example, for ListOps (Sec. 3.3), the processing should start from the deepest point in the parse tree, which is probably somewhere in the middle of the sequence. The experiments on other tasks (Sec. 3.2 and 3.3) requiring arbitrary processing orders show that bidirectional LSTMs do not generalize well in such tasks. This is not satisfactory since our goal is to create a generic architecture which can solve arbitrary problems with an arbitrary underlying input processing order. While the Transformer seems to be a good candidate for learning problem dependent processing orders, the baseline Transformer variants fail to generalize in this task in both directions.

By introducing the copy gate (Sec. 2.1), the relative Transformer can solve the forward task, but not the backward one. Our analysis showed that the network learns to attend to the last operation based on the relative position information. Since the result is read from the last column, this position changes with the sequence length. The model thus fails to generalize to such arbitrary offsets. To address this issue, we introduce a simple mechanism to let the model choose between absolute and relative positional encodings at each position (see Appendix B). The resulting model effectively manages to use the absolute position for the prediction and perform well in both directions. However, such a combination of absolute/relative positional encoding might be an overly specific bias. A more generic solution, geometric attention (Sec. 2.2), also achieved perfect generalization and was found easier to train. We present the corresponding visualization of our model in Sec. 4.

Table 1: Accuracy on **compositional table lookup** dataset.

| Model | IID | | Longer | |
|---|---|---|---|---|
| | Forward | Backward | Forward | Backward |
| LSTM | **1.00 ± 0.00** | 0.59 ± 0.03 | **1.00 ± 0.00** | 0.22 ± 0.03 |
| Bidirectional LSTM | **1.00 ± 0.00** | **1.00 ± 0.00** | **1.00 ± 0.00** | **1.00 ± 0.00** |
| DNC | **1.00 ± 0.00** | 0.57 ± 0.06 | **1.00 ± 0.00** | 0.18 ± 0.02 |
| Transformer | **1.00 ± 0.00** | 0.82 ± 0.39 | 0.13 ± 0.01 | 0.12 ± 0.01 |
| + rel | **1.00 ± 0.00** | **1.00 ± 0.00** | 0.23 ± 0.05 | 0.13 ± 0.01 |
| + rel + gate | **1.00 ± 0.00** | **1.00 ± 0.00** | **0.99 ± 0.01** | 0.19 ± 0.04 |
| + abs/rel + gate | **1.00 ± 0.00** | **1.00 ± 0.00** | 0.98 ± 0.02 | **0.98 ± 0.03** |
| + geom. att. | 0.96 ± 0.04 | 0.93 ± 0.06 | 0.16 ± 0.02 | 0.15 ± 0.02 |
| + geom. att. + gate (NDR) | **1.00 ± 0.00** | **1.00 ± 0.00** | **1.00 ± 0.00** | **1.00 ± 0.00** |

## 3.2 SIMPLE ARITHMETIC

In order to validate the success of the proposed model on a task that involves more complex data flows and operations, we propose the *simple arithmetic* task.

**Task.** The task is to execute an arithmetic expression consisting of nested modulo 10 additions and multiplications. This requires the model to process tree-structured data flows, which is presumably more difficult than the sequential processing required for the CTL task. Each operation is surrounded by brackets, such that the boundaries of operations are easy to determine. For example '((4*7)+2)' should evaluate to '0' (30 modulo 10). The expressions are generated randomly. The tree depth is up to 5 for the training set, 6 for the validation set, and 7-8 for the test set. The depth is measured as the number of operations, ignoring the leaves, so the example above has a depth of 2. The sequence length is limited to at most 50 tokens.

**Results.** Table 2 shows the results. All considered models perform well on the IID validation data, but none except the NDR performs well on the generalization test set, which achieves near-perfect accuracy of 98%. We also note that the NDR learns very quickly: while all other models require about 200 K steps to converge, the NDR achieves near-perfect accuracy after 50 K steps of training.

Table 2: Performance of different models on the **simple arithmetic** dataset. All models are trained for 200 K iterations, except the NDR which we stop training at 100 K. We also report the performance after 50 K iterations, where it can be seen that NDR converges significantly faster than the others.

| | IID (1..5) | Test (7..8) | |
| --- | --- | --- | --- |
| | 200 K | 200 K | 50 K |
| LSTM | **0.99 ± 0.00** | 0.74 ± 0.02 | 0.72 ± 0.01 |
| Bidirectional LSTM | **0.98 ± 0.01** | 0.82 ± 0.06 | 0.80 ± 0.04 |
| Transformer | **0.98 ± 0.01** | 0.47 ± 0.01 | 0.29 ± 0.01 |
| + rel | **1.00 ± 0.00** | 0.77 ± 0.04 | 0.40 ± 0.05 |
| + abs/rel + gate | **1.00 ± 0.01** | 0.80 ± 0.16 | 0.73 ± 0.15 |
| + geom. att. + gate (NDR) | **1.00 ± 0.00** | **0.98 ± 0.01** | **0.98 ± 0.01** |

## 3.3 LISTOPS

We also evaluate our model on a variant of the ListOps task (Nangia & Bowman, 2018) which is a popular task commonly used to evaluate parsing abilities of NNs (Havrylov et al., 2019; Shen et al., 2019; Xiong et al., 2021; Tay et al., 2021; Irie et al., 2021). Some special architectures such as Chowdhury & Caragea (2021) can almost perfectly generalize to longer sequences on this task. However, as far as we know, no Transformer variant has been reported to be fully successful.

**Task.** The task consists of executing nested list operations written in prefix notation. All operations have a list of arguments that can be either a digit (from 0 to 9) or recursively another operation with its own list of arguments. The operations are min, max, median and sum. The sum is modulo 10, and the median is followed by the floor function such that the output of any operation lies between 0 and 9. For example: `[MED 4 8 5 [MAX 8 4 9 ] ]` should return 6. There are two well-known variants: the original one by Nangia & Bowman (2018) and the "Long Range Arena" variant by Tay et al. (2021) which have different maximum numbers of arguments in each function and maximum sequence lengths. In both variants, there is no strict control of the depth of data samples: there is simply a certain pre-defined probability that each argument in the list is expanded into another list (which may increase the tree depth). This is not suitable for evaluating systematic generalization in terms of compositionality (over the problem depth). We propose instead to generate clean train, valid, and test splits with disjoint depths: up to depth 5 for training, depth 6 for validation and depths 7 and 8 for test. Importantly, we make sure that a depth-$K$ sample effectively requires computation until depth-$K$ (otherwise min, max, and med operations could potentially find the output without executing all of its arguments). By dissociating the splits by the depth, we can clearly identify models which fail to generalize compositionally. Apart from the depth specifications, all train/valid/test sets share the same settings as follows: the maximum sequence length is 50 (tokens), the probability of recursively sampling another function inside a list is 30% at each position, and the maximum number of arguments for a function is 5. The train set consists of 1M, the valid and test sets of 1K sequences.

**Results.** Table 3 shows the results. Like on the other tasks, the baseline LSTM and Transformers do not generalize well on the test set consisting of deeper problems, while they achieve a near-perfect accuracy on IID data. In contrast, our model achieves near-perfect generalization.

Table 3: Performance of different models on balanced **ListOps** dataset. All models are trained for 200 K iterations, except all `+gate` variants which converge after 100 K steps. The numbers in the parentheses indicate the problem depths (1-5 for the IID, and 7-8 for the test set).

| | IID (1..5) | Test (7..8) |
| --- | --- | --- |
| LSTM | **0.99 ± 0.00** | 0.71 ± 0.03 |
| Bidirectional LSTM | **1.00 ± 0.00** | 0.57 ± 0.04 |
| Transformer | 0.98 ± 0.00 | 0.74 ± 0.03 |
| + rel | **0.98 ± 0.01** | 0.79 ± 0.04 |
| + abs/rel + gate | **1.00 ± 0.01** | 0.90 ± 0.06 |
| + geom. att. + gate (NDR) | **1.00 ± 0.00** | **0.99 ± 0.01** |

## 4 ANALYSIS

In this section, we provide some visualizations of attention and gating patterns of the NDR and the corresponding analyses. For more visualizations, we refer the readers to Appendix D.

**Compositional Table Lookup.** Figure 2 shows the gating and attention patterns of the NDR model for an example of the backward presentation task. As shown in Fig. 2/Bottom, the gates of different columns open sequentially one after another when the input is available for them. Fig. 2/Top shows the corresponding attention maps. Each column attends to the neighbouring one, waiting for its computation to be finished. The behavior of the last column is different: it always attends to the second position of the sequence, which corresponds to the last operation to be performed.

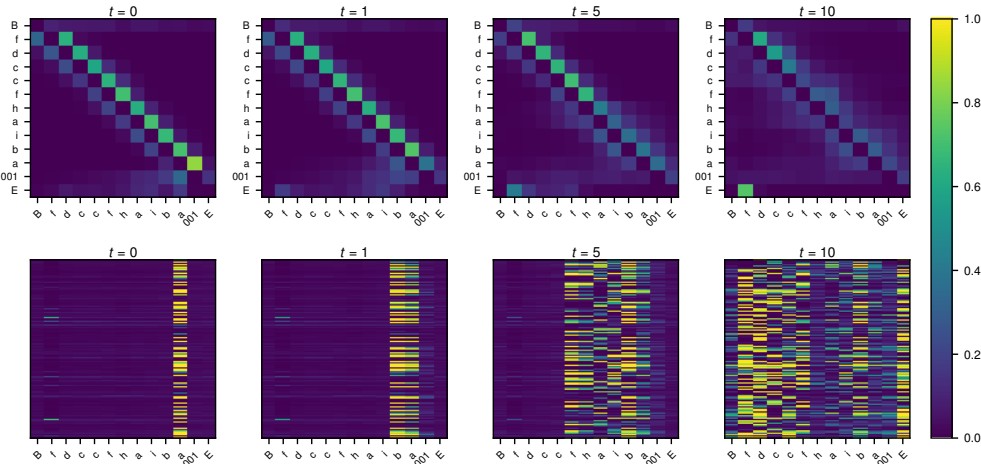

Figure 2: Example visualization of NDR. For other models, see Appendix D. Top: Attention map for different steps. The x/y-axis corresponds to source/target positions, respectively. Each position focuses on the column to the right, except the last one where the result is read from, which focuses on the last operation. The focus becomes clear only once the result is available. Bottom: gate activations for different steps/layers. The gates remain closed until the data dependencies are satisfied.

**ListOps.** We can also identify how the NDR processes the data in ListOps. Different attention heads play different roles. We highlight the core observations in Figure 3. The input for this example is: `[SM [MED [MIN 1 7 4 [MAX 2 4 0 8 9 ] ] 7 ] 5 [MED 8 5 8 ] 0 7 ]`. First of all, we find that there is a head (*head 13* in Figure 3, *first row*) which seems to be responsible for connecting operators and their arguments: the operands/arguments of an operation attend to the operator. In step 0 ($t = 0$ in the figure), we can recognize that the operations at the deepest level, namely `MAX` and the second `MED` have all the arguments ready (as is shown by vertical lines on the columns corresponding to `MAX` and `MED`). The model indeed identifies that these two operations are ready to be executed and that they can be processed in parallel (these arguments-to-operation attention patterns remain for a few steps). We note that at this stage, the last argument of `MIN` is not ready yet (`[MIN 1 7 4 [MAX 2 4 0 8 9 ] ]`). We can see that only arguments which are already ready (`1 7 4`) attend to the operator (see the column of `MIN`). In step 1 ($t = 1$, *2nd row*), we can see that *head 5* copies the expected result of `MAX`, `9` to the column of the operator (we note that this only requires one step as `9` is always the result of `MAX` when it is one of the arguments of `MAX`). Similarly in step 2, *head 7* (*2nd row*) seems to copy the result of the second `MED`, `8` to the operator column. In step 3 ($t = 3$, *1st row*), we recognize that the result of `MAX` is marked as an argument for `MIN` in *head 13* which is responsible for communication between operators and their arguments. This is shown by the new attention which appears at $t = 3$ in *head 13* from the source position `MAX` to the target position `MIN` (a pattern which is not visible at $t = 2$). In *head 3*, $t = 6$ (*2nd row*), the expected result of `MIN`, which is `1`, is copied to the operator, similarly to the patterns we observed above for `MAX` and `MED`. In *head 13*, $t = 6$ (*1st row*), all arguments for the first `MED` are now also recognized (the result of `MIN` which is `1`, and `7`). Finally in $t = 7$ (*2nd row*), two heads, *head 3* and *head 5* seem to copy/gather two inputs needed to compute the corresponding median, `1` and `7`, and store them in

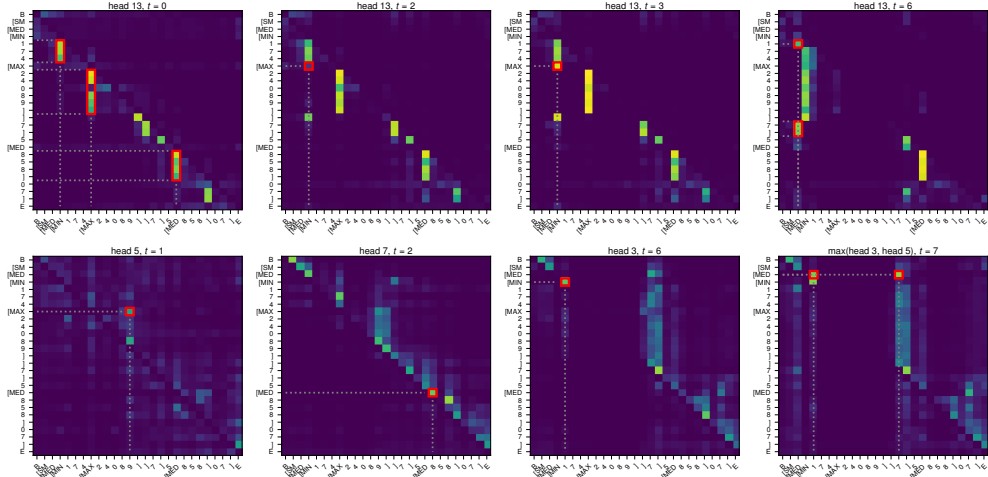

Figure 3: Example visualization of NDR on ListOps. The top row shows *head 13* in different steps, which controls which arguments are used in which step. The bottom row shows different heads in different key steps. Please refer to Sec. 4 for the step-by-step description. More visualizations are provided in the appendix: Fig. 12 shows the max of attention over all heads for all steps, Fig. 13 shows all steps of *head 13*, and Fig. 14 shows the corresponding gates.

the column of the operator MED. A complete visualization of further steps can be found in Appendix D.2. We noticed that some of the heads do not seem to play a key role; we focused on interpreting those which seem to participate in the main computation. For ListOps, we also partially find the attention patterns described above in the baseline Transformer with relative positional encoding, at least on some inspected examples, which also explains its rather high accuracy.

## 5 DISCUSSION

**Learning adaptive serialization.** The NDR architecture can be understood as performing adaptive serialization of the problem. A key requirement for reusable computation is decomposing the problem into reusable building blocks, typically applied in sequential steps. The granularity of the decomposition determines the degree of reusability: fusing operations in a single step makes the processing faster (fewer steps), but also more specialized. Learning the most granular solutions is thus preferable for generalization. At the same time, not all processing should happen serially: branches of the computational graph that do not have common data dependencies can be processed independently in parallel, which we empirically observe in our NDR in the ListOps example (Sec. 4). This enables the architecture to get away with a number of computational steps reflecting the depth of the computational graph rather than the length of the input.

**Bottom up approach for improving model architectures.** Transformers have seen tremendous successes across various application domains (Devlin et al., 2019; Brown et al., 2020; Dosovitskiy et al., 2021). Impressive results have been reported when they are scaled up with a large amount of data (Brown et al., 2020). On the other hand, simple tasks like those highlighted in the present work demonstrate that the Transformer architecture still struggles with basic reasoning. Particularly in algorithmic tasks, it is often the case that a sub-optimal choice of architecture/optimization method makes the model fall back to simple memorization. We argue that it is crucial to look at isolated problems which test specific generalization capability. This calls for a bottom-up approach: building on toy tasks that focus on individual aspects of generalization and using them for improving models.

## 6 RELATED WORK

**Gating inside Transformers.** Several prior works have proposed to use some sort of gating within Transformer architectures (Parisotto et al., 2020; Chaabouni et al., 2021). Our proposed *copy gate*

is different from those as it satisfies two important properties. First, our copy gate allows the model to skip the *entire* Transformer layer (i.e., both the self-attention and the feedforward blocks) when the gate is closed. Second, the gate function is conditioned on the attention output such that the decision of opening or closing depends on information from all columns. While multiple gating variants have been proposed by Parisotto et al. (2020) to stabilize Transformers for reinforcement learning, none of them can produce this behavior. Empirically, we also tried out a few other gating variants which do not satisfy the two properties above; we found them not to improve over regular Transformers in our preliminary experiments on compositional table lookup. Recent work by Chaabouni et al. (2021) also makes use of "gating" in Transformers through a gated linear unit (GLU) activation function commonly used in convolutional NNs (Dauphin et al., 2017). Transformer models with such an activation function were reported to outperform RNN baselines on a systematic generalization task (Dessì & Baroni, 2019). Unlike our copy gate or Parisotto et al. (2020)'s gating, such a gating activation does not have the "residual" term (i.e. a closed gate zeros out the input) which allows the model to skip a transformation. In a more general context, benefits of the GLU activation in Transformers vary across tasks (Irie et al., 2019; Shazeer, 2020). In language modeling, no improvement is typically obtained by using the standard highway gate instead of the residual connection in Transformers (Irie, 2020), while it yields improvements when combined with convolutional layers (Kim & Rush, 2016).

**Parameterized geometric distributions.** Two recent works (Brooks et al., 2021; Banino et al., 2021) have used a form of parameterized geometric distribution (PGD; in the form of Eq. 7). Brooks et al. (2021) have used such a distribution to parameterize the movement of a pointer on a sequence of instructions. Banino et al. (2021) have used it to implement adaptive computation time (Schmidhuber, 2012; Graves, 2016). We use the PGD to obtain a generic attention mechanism as a replacement of the standard self-attention used in Transformers (Vaswani et al., 2017).

**Compositional table lookup.** CTL task was proposed for evaluating the compositional ability of NNs (Liska et al., 2018). Previous works evaluated RNNs, RNNs with attention, and Transformers on this task with limited success (Hupkes et al., 2019; Dubois et al., 2020). Dubois et al. (2020) have proposed a special attention mechanism to augment the recurrent architecture. While they obtained good performance for the forward presentation order, the proposed model failed in the backward one. In contrast, two of our approaches (Sec. 3.1) achieve 100% generalization accuracy for both orders.

**Positional encodings.** Many previous works have focused on improving positional encoding (Schmidhuber, 1992; Vaswani et al., 2017) for self-attention. Most notably, the relative positional encoding (Schmidhuber, 1992; Shaw et al., 2018; Dai et al., 2019) was found useful for improving systematic generalization of Transformers (Csordás et al., 2021). Here we also present two new approaches related to positional encoding. One is the gated combination of absolute and relative positional encoding (Sec. 3.1; details in Appendix B). We show that absolute positional encoding can complement relative positional encoding. The former enables the model to always attend to a specific position, as is needed for the CTL task in the last step, while the gating allows it to use relative positional encoding for other positions/steps. Second, we introduce directional encoding to augment geometric attention. Unlike positional encoding which can overfit to a range of positions seen during training, the direction information is found to be robust and to be a crucial augmentation of the geometric attention.

## 7 CONCLUSION

We proposed a new view on the internal operations of Transformer encoders as a dynamic dataflow architecture between Transformer columns. This overcomes two shortcomings of traditional Transformers: the problem of routing and retaining data in an unaltered fashion, which we solve by an additional copy gate, and the problem of learning length-independent attention patterns, which we solve by geometric attention. Our new model, the Neural Data Router (NDR), generalizes to compositions longer than those seen during training on the popular compositional lookup table task in both forward and backward directions. NDR also achieves near perfect performance on simple arithmetic and ListOps tasks in settings that test systematic generalization in terms of computational depth. In general, the gates and the attention maps collectively make the architecture more interpretable than the baselines. Future work will extend this encoder-only architecture to a full sequence-to-sequence model and evaluate it on other standard tasks in systematic generalization requiring generation of variable-length output sequences.

## ACKNOWLEDGMENTS

We thank Imanol Schlag and Sjoerd van Steenkiste for helpful discussions and suggestions on an earlier version of the manuscript. This research was partially funded by ERC Advanced grant no: 742870, project AlgoRNN, and by Swiss National Science Foundation grant no: 200021_192356, project NEUSYM. We are thankful for hardware donations from NVIDIA & IBM. The resources used for the project were partially provided by Swiss National Supercomputing Centre (CSCS) project s1023.

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

## A ABLATIONS

**Number of layers.** In Sec. 2, we hypothesized that decomposition of the problem into its elementary operations is a necessary property of a model which generalizes. This motivated us to configure our models to have at least as many layers as the depth of the computation involved, plus a few additional layers for writing the output and for gathering an overview of the problem at the beginning. We assumed that in such a model with a sufficient number of layers, each layer learns the underlying "elementary" operation. The resulting models are thus deeper than those typically used in the literature for similar tasks (Keysers et al., 2020; Tay et al., 2021). Here we provide an ablation study to demonstrate that such depths are effectively necessary for generalization. We measure the IID and generalization performance with various numbers of layers on the compositional table lookup dataset. Since our test set on the CTL task consists of up to 10 function applications, it should require about 12 layers according to our hypothesis. Table 4 shows the results. We clearly observe that, while the shallow models also solve the IID split, only the deep models generalize to the longer problems (here the 12-layer model generalizes almost perfectly, but the 10-layer one does not).

Table 4: The performance of NDR on the **compositional table lookup** dataset, with different number of layers.

| $n_{\text{layers}}$ | IID | | Test | |
|---|---|---|---|---|
| | Forward | Backward | Forward | Backward |
| 14 | $1.00 \pm 0.00$ | $1.00 \pm 0.00$ | $\mathbf{1.00 \pm 0.00}$ | $\mathbf{1.00 \pm 0.00}$ |
| 12 | $1.00 \pm 0.00$ | $1.00 \pm 0.00$ | $\mathbf{1.00 \pm 0.00}$ | $\mathbf{0.99 \pm 0.02}$ |
| 10 | $1.00 \pm 0.00$ | $1.00 \pm 0.00$ | $0.75 \pm 0.04$ | $0.62 \pm 0.05$ |
| 8 | $1.00 \pm 0.00$ | $1.00 \pm 0.00$ | $0.23 \pm 0.02$ | $0.24 \pm 0.03$ |
| 6 | $1.00 \pm 0.00$ | $0.96 \pm 0.03$ | $0.22 \pm 0.05$ | $0.15 \pm 0.01$ |
| 4 | $0.96 \pm 0.04$ | $0.68 \pm 0.11$ | $0.14 \pm 0.01$ | $0.13 \pm 0.01$ |

**Readout from the first instead of the last column.** In our experiments with the Transformer models, the last column was used for the readout of the result. Under this configuration, the readout position depends on the length of the sequence which might increase the difficulty of the problem, in particular for the models using absolute positional embeddings. Table 5 shows the corresponding ablation study. We observe that this choice has only marginal impact on the model performance. As a side note, we also tried the variant where an additional cross-attention layer is used for the readout. Again, the generalization performance was not better. In fact, these results are not surprising since none of these changes fundamentally addresses the problem of length generalization.

Table 5: Accuracy on **compositional table lookup** dataset with the results read from the first or last column (**Readout**).

| Model | Readout | IID | | Longer | |
|---|---|---|---|---|---|
| | | Forward | Backward | Forward | Backward |
| Transformer | First | $1.00 \pm 0.00$ | $0.82 \pm 0.39$ | $0.12 \pm 0.01$ | $0.13 \pm 0.01$ |
| | Last | $1.00 \pm 0.00$ | $0.82 \pm 0.39$ | $0.13 \pm 0.01$ | $0.12 \pm 0.01$ |
| + rel | First | $1.00 \pm 0.00$ | $1.00 \pm 0.00$ | $0.12 \pm 0.01$ | $0.22 \pm 0.05$ |
| | Last | $1.00 \pm 0.00$ | $1.00 \pm 0.00$ | $0.23 \pm 0.05$ | $0.13 \pm 0.01$ |
| + rel + gate | First | $1.00 \pm 0.00$ | $1.00 \pm 0.00$ | $0.17 \pm 0.02$ | $1.00 \pm 0.00$ |
| | Last | $1.00 \pm 0.00$ | $1.00 \pm 0.00$ | $0.99 \pm 0.01$ | $0.19 \pm 0.04$ |

**Does Adaptive Computation Time (ACT) help?** In this work, we determined the number of layers/steps to be used in the model based on heuristics (see Appendix C.1). We could also consider using Adaptive Computation Time (ACT) to dynamically determine the number of steps. Furthermore, ACT introduces a form of gating which creates shortcuts in the credit assignment path between the output and a result of an intermediate layer. This "copying" mechanism resulting from the ACT (i.e. stop computation at a certain time and copy the result to the output) is fundamentally different

from our copy gate (Sec. 2.1). Our copy gate allows Transformer columns to keep the input unchanged until it's their turn to be processed (a crucial property to implement control flow like behavior). This behavior can not be simulated by the ACT. Here we provide some experimental results on models with ACT which confirm that the proposed copy gate is a crucial component for generalization which can not be replaced by ACT.

We note that there are various versions of ACT in the literature, e.g., the variant used by Dehghani et al. (2019) in Universal Transformers is different from the one used by Graves (2016). Here we focus on two variants: one in which we directly apply Graves (2016) to Transformers, and another one used by Dehghani et al. (2019). We start with the description of the former.

An extra sigmoidal unit $\hat{p}^{(i,t)}$ is computed for each column $i$ in each timestep $t$ as:

$$\hat{p}^{(i,t)} = \sigma(\boldsymbol{W}_{\mathrm{H}}\boldsymbol{h}^{(i,t)} + b_{\mathrm{H}}) \tag{11}$$

where $\boldsymbol{W}_{\mathrm{H}} \in \mathbb{R}^{1 \times d}$ and $\boldsymbol{b}_{\mathrm{H}} \in \mathbb{R}$ are trainable parameters. By comparing the cumulative sum of $\hat{p}^{(i,t)}$ over time steps to a certain threshold value $(1 - \epsilon)$ with a hyper-parameter $\epsilon$ (0.01 in our experiment), we determine the termination step $T^i$ for column $i$ as:

$$T^i = \min\{T_{\mathrm{max}}, \min\{t' : \sum_{t=1}^{t'} \hat{p}^{(i,t)} \geq 1 - \epsilon\}\} \tag{12}$$

where $T_{\mathrm{max}}$ is the pre-defined maximum number of steps.

The corresponding *halting probability* $p^{(i,t)}$ is then computed as:

$$p^{(i,t)} = \begin{cases} \hat{p}^{(i,t)} & \text{if } t < T^i \\ R^i & \text{if } t = T^i \end{cases} \tag{13}$$

$$R^i = 1 - \sum_{t=1}^{T^i-1} \hat{p}^{(i,t)} \tag{14}$$

which is used to compute the final output of column $i$ as:

$$\boldsymbol{o}^i = \sum_{t=1}^{T^i} p^{(i,t)} \boldsymbol{h}^{(i,t)} \tag{15}$$

In Dehghani et al. (2019)'s variant, a different equation is used in lieu of Eq. 15 above and the computation of the reminder term $R^i$ in Eq. 14 above is not properly handled in case where Eq. 12 terminates because of the first condition on $T_{\mathrm{max}}$. For further details, we refer the readers to Listing 1 and 2 in Dehghani et al. (2019) and/or our public code.

One subtlety introduced by Dehghani et al. (2019) which we note here is that the computation of the final output $\boldsymbol{o}^i$ of column $i$ effectively "halts" after $T^i$ (since $\boldsymbol{o}^i$ only depends on $\boldsymbol{h}^{(i,t)}$ for $0 < t < T^i$), but column $i$ itself still continues transforming the hidden states $\boldsymbol{h}^{(i,t)}$ for steps $t > T^i$ until all columns reach the termination step, and its updated states can be attended/read by another column $j$ which has not halted yet (i.e. $T^j > T^i$). In this sense, computation is never stopped independently for each column. The mechanism described above instead finds the *readout* steps for each column (as is used in Eq. 15). We follow this decision in our implementation of both variants.

In addition, a new regularizer term, $L_{\mathrm{ACT}} = \alpha \frac{1}{N} \sum_{i=1}^{N} R^i$ is added to the loss function, where $N$ is the length of the input sequence. This makes the network prefer short computations. We ran a hyper-parameter search for $\alpha$ from the following values: 0.001, 0.003, 0.01, 0.03, 0.1. We found $\alpha = 0.03$ to work the best.

We conducted experiments on the compositional table lookup task. We first noted that ACT helps training our baseline Transformer models with a maximum step of 14 layers which was not possible without ACT (our baseline Transformer had only 11 layers for this reason; see Table 7). The shortcut in the credit assignment path introduced by ACT certainly helps training of this 14 layer model. As we noticed that the models with ACT learn slower than those with gating, we increased the number of training steps to 60k steps which is twice as many as 30k used for the models without ACT. Table

6 shows the results. We observe that, interestingly, ACT enables generalization for longer lengths in the forward direction of the Transformer with relative positional encoding and the one with geometric attention. However, we were not able to find any configuration that generalizes in the backward case. This demonstrates that the copy gate is effectively a crucial component for generalization which can not be replaced by ACT. Furthermore, the convergence of models with ACT is significantly slower than those of models with our gating, and they are more unstable and very sensitive to the value of $\alpha$ on the regularization term, even in the successful forward case. Overall, the only benefit of ACT is thus the adaptive depth, as is illustrated in Figure 4, which is orthogonal to our study.

Table 6: Accuracy on **compositional table lookup** dataset with adaptive computation time (ACT). Two variants of ACT are shown: "U" corresponds to Dehghani et al. (2019), while "A" is the variant described in Appendix A. We also include baselines without ACT from Table 1 as a reference. Generalization performance after 30k and 60k **training steps** are shown.

| Model | ACT | IID | | Longer, 60k | | Longer, 30k | |
|---|---|---|---|---|---|---|---|
| | | Forward | Backward | Forward | Backward | Forward | Backward |
| Transformer | | $1.00 \pm 0.00$ | $0.82 \pm 0.39$ | - | - | $0.13 \pm 0.01$ | $0.12 \pm 0.01$ |
| | A | $1.00 \pm 0.00$ | $1.00 \pm 0.00$ | $0.12 \pm 0.02$ | $0.12 \pm 0.01$ | $0.13 \pm 0.01$ | $0.13 \pm 0.01$ |
| | U | $1.00 \pm 0.00$ | $1.00 \pm 0.00$ | $0.12 \pm 0.01$ | $0.11 \pm 0.01$ | $0.13 \pm 0.01$ | $0.12 \pm 0.01$ |
| + rel | | $1.00 \pm 0.00$ | $1.00 \pm 0.00$ | - | - | $0.23 \pm 0.05$ | $0.13 \pm 0.01$ |
| | A | $1.00 \pm 0.00$ | $1.00 \pm 0.00$ | $0.99 \pm 0.02$ | $0.13 \pm 0.00$ | $0.84 \pm 0.22$ | $0.13 \pm 0.01$ |
| | U | $1.00 \pm 0.00$ | $1.00 \pm 0.00$ | $0.92 \pm 0.14$ | $0.12 \pm 0.02$ | $0.67 \pm 0.41$ | $0.12 \pm 0.00$ |
| + geo | | $0.96 \pm 0.04$ | $0.93 \pm 0.06$ | - | - | $0.16 \pm 0.02$ | $0.15 \pm 0.02$ |
| | A | $1.00 \pm 0.00$ | $1.00 \pm 0.00$ | $0.97 \pm 0.05$ | $0.45 \pm 0.21$ | $0.58 \pm 0.16$ | $0.30 \pm 0.17$ |
| | U | $0.96 \pm 0.10$ | $1.00 \pm 0.00$ | $0.72 \pm 0.35$ | $0.44 \pm 0.19$ | $0.31 \pm 0.22$ | $0.21 \pm 0.07$ |
| + rel + gate | | $1.00 \pm 0.00$ | $1.00 \pm 0.00$ | - | - | $\mathbf{0.99 \pm 0.01}$ | $0.19 \pm 0.04$ |
| + abs/rel + gate | | $1.00 \pm 0.00$ | $1.00 \pm 0.00$ | - | - | $\mathbf{0.98 \pm 0.02}$ | $\mathbf{0.98 \pm 0.03}$ |
| + geo + gate (NDR) | | $1.00 \pm 0.00$ | $1.00 \pm 0.00$ | - | - | $\mathbf{1.00 \pm 0.00}$ | $\mathbf{1.00 \pm 0.00}$ |

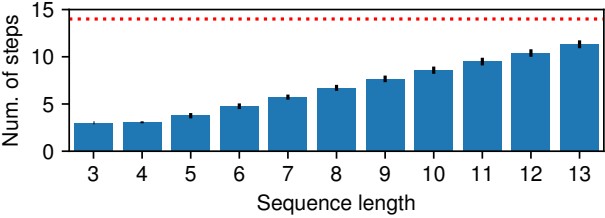

Figure 4: Average number of steps/layers for different sequence lengths on the compositional table lookup task for the Transformer with relative positional encodings and the ACT variant described in Appendix A. The red line shows $T_{\max} = 14$. Note that the sequence length shown here includes the begin and end tokens. Thus, the sequence length of 4 corresponds to one function application (3 for the identity function i.e. no function is applied).

# B DETAILS OF ATTENTION WITH COMBINED ABSOLUTE/RELATIVE POSITIONAL ENCODING

The use of copy gates enables Transformers to generalize to longer lengths in the forward presentation order of the CTL task (Sec. 3.1), but that alone was not enough to make the model generalize in the backward order variant of the task. Examining the attention maps reveals that the model uses position-based attention to read out the result instead of content-based attention. In the backward presentation order, the last column of the transformer should focus on the second column, whose relative position changes dynamically with the length of the sequence. We solve this issue by adding an option to choose between absolute and relative positional encodings to the attention head.

In what follows, we describe the operation within a single layer/step. This allows us to omit the layer/step-index $t$ for better readability, and thus denote the state of column/position $i$ as $\boldsymbol{h}_i$ instead of $\boldsymbol{h}^{(i,t)}$. We use the relative positional embedding variant of self-attention by Dai et al. (2019). Our attention matrix with the gated absolute/relative positional encodings can be decomposed as follows:

$$r_i = \sigma(\boldsymbol{h}_i \boldsymbol{W}_{ar} + b_{ar}) \tag{16}$$

$$\hat{\boldsymbol{A}}_{i,j} = \underbrace{\boldsymbol{h}_i^\top \boldsymbol{W}_q^\top \boldsymbol{W}_{k,E} \boldsymbol{h}_j}_{(a)} + \underbrace{\boldsymbol{b}_{q,E}^\top \boldsymbol{W}_{k,E} \boldsymbol{h}_j}_{(c)} + \left( \underbrace{\boldsymbol{h}_i^\top \boldsymbol{W}_q^\top \boldsymbol{W}_{k,P}}_{(b)} + \underbrace{\boldsymbol{b}_{q,P}^\top \boldsymbol{W}_{k,P}}_{(d)} \right) \underbrace{\left( \boldsymbol{p}_{i-j} r_i + \boldsymbol{p}_j (1 - r_i) \right)}_{(e)}$$

$$\tag{17}$$

where the matrix $\boldsymbol{W}_q \in \mathbb{R}^{d_{\text{head}} \times d}$ maps the states to queries, $\boldsymbol{W}_{k,E} \in \mathbb{R}^{d_{\text{head}} \times d}$ maps states to keys, while $\boldsymbol{W}_{k,P} \in \mathbb{R}^{d_{\text{head}} \times d}$ maps positional embeddings to keys. $d_{\text{head}}$ is the size of the key, query and value vectors for each head, set as $d_{\text{head}} = \frac{d}{n_{\text{head}}}$. $\boldsymbol{b}_{q,E}, \boldsymbol{b}_{q,P} \in \mathbb{R}^{d_{\text{head}}}$ are learned vectors. $\boldsymbol{p}_i \in \mathbb{R}^d$ is the standard sinusoidal embedding for position $i$ (Vaswani et al., 2017). Softmax is applied to the second dimension of $\hat{\boldsymbol{A}}$ to obtain the final attention scores, $\boldsymbol{A}$. Component (a) corresponds to content-based addressing, (b, e) to content-based positional addressing, (c) represents a global content bias, while (d, e) represent a global position bias.

We introduce term (e) for the positional embedding which can switch between absolute and relative positional encodings using the scalar gate $r_i$ (Eq. 16; parameterized by $\boldsymbol{W}_{ar} \in \mathbb{R}^{d \times 1}$ and $b_{ar} \in \mathbb{R}$), which is the function of the state at target position $i$.

## C  IMPLEMENTATION DETAILS

A PyTorch implementation of our models together with the experimental setup is available under `https://github.com/robertcsordas/ndr`. The performance of all models is reported as mean and standard deviations over 5 different seeds.

### C.1  CHOOSING THE NUMBER OF LAYERS

In Sec. 2, we hypothesized that one of the conditions for our model to generalize is to be "sufficiently" deep such that elementary operations are learned in separate layers which would then become composable. In practice, a "sufficient" depth can be determined by the basic units of compositions implicitly defined by the dataset. The depth of the model must be at least as deep as the deepest path in the computation graph defined by these basic operations. This hypothesis was empirically validated in the ablation study presented above (Appendix A). In general, we used the following heuristics to choose the depth of the Transformers:

(length of the deepest path in the graph) $\times$ (steps per operation) + a few more layers.

Determining the number of steps needed by the elementary operation is not straightforward but it can be done empirically. For example, for ListOps, as is shown in Sec. 4, it requires two steps per operation: one step in which the operands attend to the operation, followed by another one where the result is written back to the operation. For other tasks, we found that a single step per operation was enough. Choosing more layers than needed is safe, and it can be used to determine the required number of layers, for example by looking at the gate activity. Finally, "+ a few more layers" are needed because one additional layer should be used to read out the final result, and one or a few more can be needed for communication between columns (e.g., to determine operator precedence).

Since parameters are shared across layers, we can optionally train models with a certain number of layers and increase the number of computational steps at test time. This allows us to train models using a depth which is "sufficient" to solve the training set, but increase it at test time to generalize to a test set requiring more computational steps. We did this for the ListOps experiment (Sec. 3.3): the model was trained with 20 layers and tested with 24. Our preliminary experiments confirmed that this practice has no performance penalty, while it speeds up training.

## C.2 DATASET DETAILS

**Compositional table lookup.** Our implementation uses 8 symbols as input arguments and 9 randomly sampled bijective functions denoted by lower case letters of the English alphabet. All functions are included in the train set in combination with all possible input symbols. The rest of the training set consists of random combinations of functions applied to a random symbol as an argument, up to length 5. The total size of the train set is 53,704 samples. The samples are roughly balanced such that there are similar numbers of samples for each depth. There are different validation sets: an IID set, which matches the distribution of the train set, and a depth validation, which includes samples of lengths 6, 7 and 8. The test set consists of sequences of lengths 9 and 10.

**Simple arithmetic.** The dataset is constructed by sampling random digits (0-9) and operations + (add) and ∗ (multiply). The operations are performed modulo 10. Parentheses surround the arguments of the operations. The depth of the resulting tree is computed, and rejection sampling is used to ensure that the same number of samples from each depth is present in the given split. The maximum length of samples is 50 tokens, sub-operations are sampled with probability 0.2. 100 K samples are used for training, 1 K for both test and validation sets. The train set consists of 0-5 operations, the validation set of 6 and the test set of 7 operations.

**ListOps.** Random digits are sampled from range 0-9. Operations are sample from the set sum-modulo (`SM`), which is a sum modulo 10, min (`MIN`), max (`MAX`) and median followed by the floor function (`MED`). The maximum number of arguments for each operation is 5. A sub-operation is sampled with probability 0.3. 1 M samples are used for training, 1 K for test and validation. The train set consists of 0-5 operations, 6 for the validation set, and 7 for the test set.

For each sample, we calculate a number which we call *dependency depth*. To understand it, note that MIN and MAX operations only select one of their operands, MED selects 1 or 2. In SUM, all operands are needed to perform the operation. If we construct a parse tree and prune away the branches which were *not* selected by any operation and measure the depth of such a tree, the resulting number is the dependency depth. This ensures that the deeper parts of the tree contribute to the result calculation, preventing shallow heuristics, like ignoring all branches of the tree that are too deep and still getting the correct result with a high chance. We also ensure that the number of samples is the same for all possible dependency depths in each split.

## C.3 MODEL DETAILS

We use the AdamW optimizer (Loshchilov & Hutter, 2019) for all of our models. Standard hyperparameters are listed in Tab. 7, 8 and 9. Additionally, models with gating use dropout (Hanson, 1990; Srivastava et al., 2014) applied to the content-based query and the position-query components of 0.1 for most models, except for non-gated Transformers on ListOps, where this value is 0.05. In the case of geometric attention, since the channels of the directional encoding does not have any redundancy, dropout is applied just to the content-query.

In the case of Transformers with the copy gate but without geometric attention, we use $\tanh$ instead of $\mathrm{LayerNorm}$ in Eq. 2. The Transformer/NDR layer with a copy gate is illustrated in Figure 5.

The hyperparameters of the gateless Transformers differ significantly from the gated ones. This is because they were very hard to train to achieve good performance even on the IID set, requiring extensive hyperparameter tuning. One might argue that fewer layers make them less competitive on longer sequences. However, we were unable to train them to perform well even on IID data with comparable sizes.

All Transformer variants have a begin (B) and end (E) token included in the sequence. RNNs (LSTM and DNC) have no such tokens. All Transformers are encoders only, and the results are read from the last column (corresponding to the end token).

The DNC has 21 memory cells, 4 read heads, and an LSTM controller. It contains recently introduced improvements (Csordás & Schmidhuber, 2019).

We use gradient clipping with magnitude 5 (for CTL) or 1 (for simple arithmetic and ListOps) for all of our models.

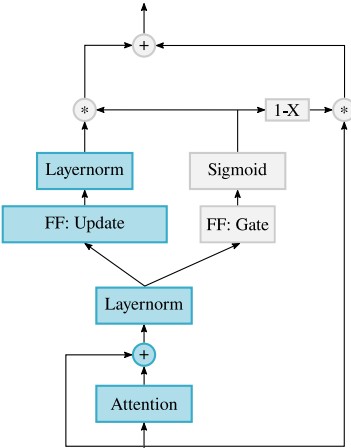

Figure 5: Structure of Transformer/NDR layer with a copy gate (Sec. 2.1). The blue part corresponds to the standard Transformer, except for the missing residual connection around the feedforward block ("FF: Update"). The gray part is the copy gate. The feedforward part corresponding to the gate is usually significantly smaller than the one used for the update.

Hyperparameters were obtained by a Bayesian hyperparameter search of Weights & Biases[2] over the systematically different (OOD) validation set for the `+abs/rel + gate` models and were reused for all other gated models. For the non-gated models, we used the `+rel` variant for tuning. It was not possible to tune the baselines using only the OOD validation set because their performance was too bad on that set. We thus used a mixture of IID and OOD validation sets to tune the hyperparameters for the baselines. Table 10 shows the range of hyperparameters used for tuning. "FF multiplier" is used to calculate $d_{FF}$ from $d_{model}$.

We train all models for a fixed number of $n_{iters}$ iterations and measure their validation performance every 1000 iterations. For each model, we select the best checkpoint according to the validation performance, and report its test accuracy.

Table 7: Hyperparameters used for different models on the compositional table lookup task. We denote the feedforward size as $d_{FF}$, weight decay as "wd.", dropout as "do.". The model is trained for $n_{iters}$ iterations.

| | $d_{model}$ | $d_{FF}$ | $n_{heads}$ | $n_{layers}$ | batch s. | learning rate | wd. | do. | $n_{iters}$ |
|---|---|---|---|---|---|---|---|---|---|
| LSTM | 200 | - | - | 1 | 256 | $10 * 10^{-4}$ | - | 0.5 | 20k |
| Bidirectional LSTM | 400 | - | - | 1 | 256 | $10 * 10^{-4}$ | - | 0.5 | 20k |
| DNC | 200 | - | - | 1 | 256 | $10 * 10^{-4}$ | - | 0.5 | 20k |
| Transformer | 128 | 256 | 4 | 11 | 512 | $1.5 * 10^{-4}$ | 0.0025 | 0.1 | 30k |
| + rel | 128 | 256 | 4 | 11 | 512 | $1.5 * 10^{-4}$ | 0.0025 | 0.1 | 30k |
| + rel + gate | 256 | 512 | 1 | 14 | 512 | $2 * 10^{-4}$ | 0.01 | 0.5 | 30k |
| + abs/rel + gate | 256 | 512 | 1 | 14 | 512 | $2 * 10^{-4}$ | 0.01 | 0.5 | 30k |
| + geom. att. | 128 | 256 | 4 | 11 | 512 | $1.5 * 10^{-4}$ | 0.0025 | 0.1 | 30k |
| + geom. att. + gate (NDR) | 256 | 512 | 1 | 14 | 512 | $1.5 * 10^{-4}$ | 0.01 | 0.5 | 30k |

# D  ADDITIONAL ANALYSIS

## D.1  COMPOSITIONAL TABLE LOOKUP

An idealized sequence of computations in a Transformer for an example from CTL task is shown in Fig. 6. Each column waits for its input from the left side, then performs an update. Finally, the last

---

[2] https://wandb.ai/

Table 8: Hyperparameters used for different models on the simple arithmetic task. We denote the feedforward size as $d_{\text{FF}}$, weight decay as "wd.", dropout as "do.". The model is trained for $n_{\text{iters}}$ iterations.

| | $d_{\text{model}}$ | $d_{\text{FF}}$ | $n_{\text{heads}}$ | $n_{\text{layers}}$ | batch s. | learning rate | wd. | do. | $n_{\text{iters}}$ |
|---|---|---|---|---|---|---|---|---|---|
| LSTM | 200 | - | - | 2 | 256 | $10 * 10^{-4}$ | - | 0.5 | 200k |
| Bidirectional LSTM | 400 | - | - | 2 | 256 | $10 * 10^{-4}$ | - | 0.5 | 200k |
| Transformer | 128 | 256 | 4 | 11 | 512 | $1.5 * 10^{-4}$ | 0.0025 | 0.5 | 200k |
| + rel | 128 | 256 | 4 | 11 | 512 | $1.5 * 10^{-4}$ | 0.0025 | 0.5 | 200k |
| + abs/rel + gate | 256 | 1024 | 4 | 15 | 512 | $1.5 * 10^{-4}$ | 0.01 | 0.5 | 100k |
| + geom. att. + gate (NDR) | 256 | 1024 | 4 | 15 | 512 | $1.5 * 10^{-4}$ | 0.01 | 0.5 | 100k |

Table 9: Hyperparameters used for different models on the ListOps task. We denote the feedforward size as $d_{\text{FF}}$, weight decay as "wd.", dropout as "do.". The model is trained for $n_{\text{iters}}$ iterations.

| | $d_{\text{model}}$ | $d_{\text{FF}}$ | $n_{\text{heads}}$ | $n_{\text{layers}}$ | batch s. | learning rate | wd. | do. | $n_{\text{iters}}$ |
|---|---|---|---|---|---|---|---|---|---|
| LSTM | 512 | - | - | 4 | 512 | $10 * 10^{-4}$ | 0.08 | 0.1 | 200k |
| Bidirectional LSTM | 1024 | - | - | 4 | 512 | $10 * 10^{-4}$ | 0.08 | 0.1 | 200k |
| Transformer | 256 | 1024 | 16 | 6 | 512 | $4 * 10^{-4}$ | 0.05 | 0.015 | 200k |
| + rel | 256 | 1024 | 16 | 6 | 512 | $4 * 10^{-4}$ | 0.05 | 0.015 | 200k |
| + abs/rel + gate | 512 | 1024 | 16 | 20 | 512 | $2 * 10^{-4}$ | 0.09 | 0.1 | 100k |
| + geom. att. + gate (NDR) | 512 | 1024 | 16 | 20 | 512 | $2 * 10^{-4}$ | 0.09 | 0.1 | 100k |

column copies the result. So far, in the main text, we only had space to show the gate and attention activity of the NDR for a few timesteps. Here we show the corresponding visualization of all steps in Figures 10 and 11, as well as the attention map for the baseline Transformer with relative positional encoding in Figure 7. We also show the `Transformer + abs/rel + gate` variant in Fig. 8 and Fig. 9. Please directly refer to the caption of the figures for the corresponding analysis. In general, the visualization for our NDR and the `abs/rel + gate` variant is easily interpretable, unlike that of the baseline Transformer model.

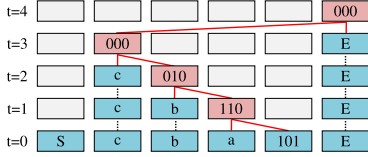

Figure 6: An ideal sequence of computations in a Transformer for an example CTL task.

## D.2 LISTOPS

Figures 12 and 14 shows the attention and gate patterns of our NDR architecture on an example from the ListOps dataset. We highlighted notable attention patterns in Sec. 4.

Different heads seem to specialize in different functions. As already mentioned in Sec. 4, *head 13* of the NDR architecture, shown in Figure 13, seems to specialize in selecting which arguments belong to which operator.

The gating patterns are also very interesting. In the early stages, the deepest parts of the input are updated: `[MAX 2 4 0 8 9]` and `[MED 8 5 8]`, which are independent branches of the parse tree that can be processed in parallel. In the following steps, the update patterns spread up in the parse tree, updating the operations that have their arguments available. In this task, the input is read from the first column, which is written in a very late stage.

Table 10: Parameter ranges for hyperparameter tuning

| Parameter | Range |
|---|---|
| learning rate | 0.00005 ... 0.001 |
| $n_{\text{layers}}$ | 4 ... 20 |
| $d_{\text{model}}$ | 128, 256, 512 |
| $n_{\text{heads}}$ | 2, 4, 8, 16 |
| weight decay | 0.0 ... 0.1 |
| dropout | 0.0 ... 0.5 |
| attention dropout | 0.0 ... 0.5 |
| FF multiplier | 1, 2, 4 |

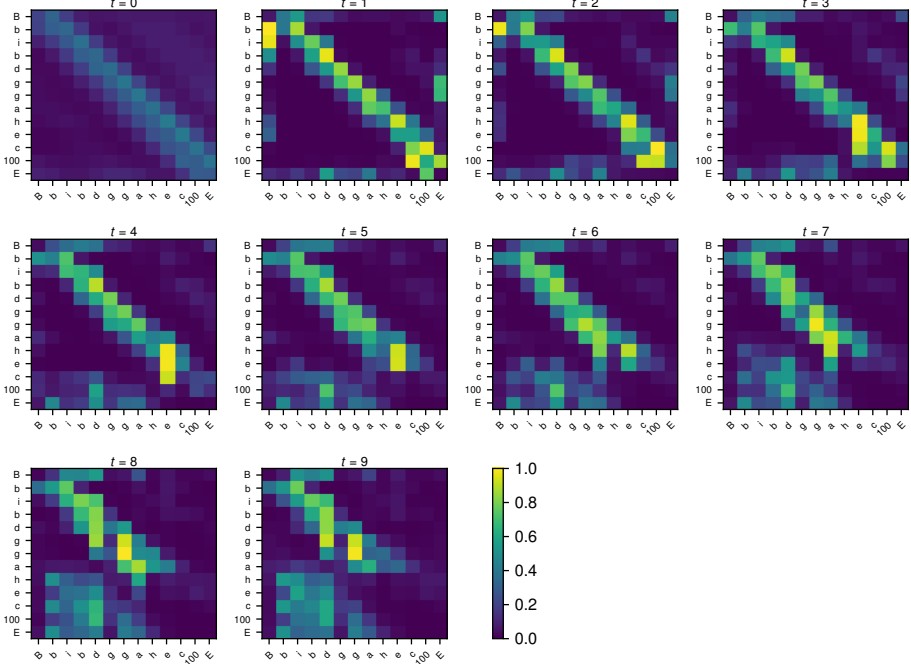

Figure 7: Attention map for every computational step for a baseline Transformer with relative positional encoding on CTL. The attention pattern gets blurry very quickly, and the model does not generalize to longer sequences.

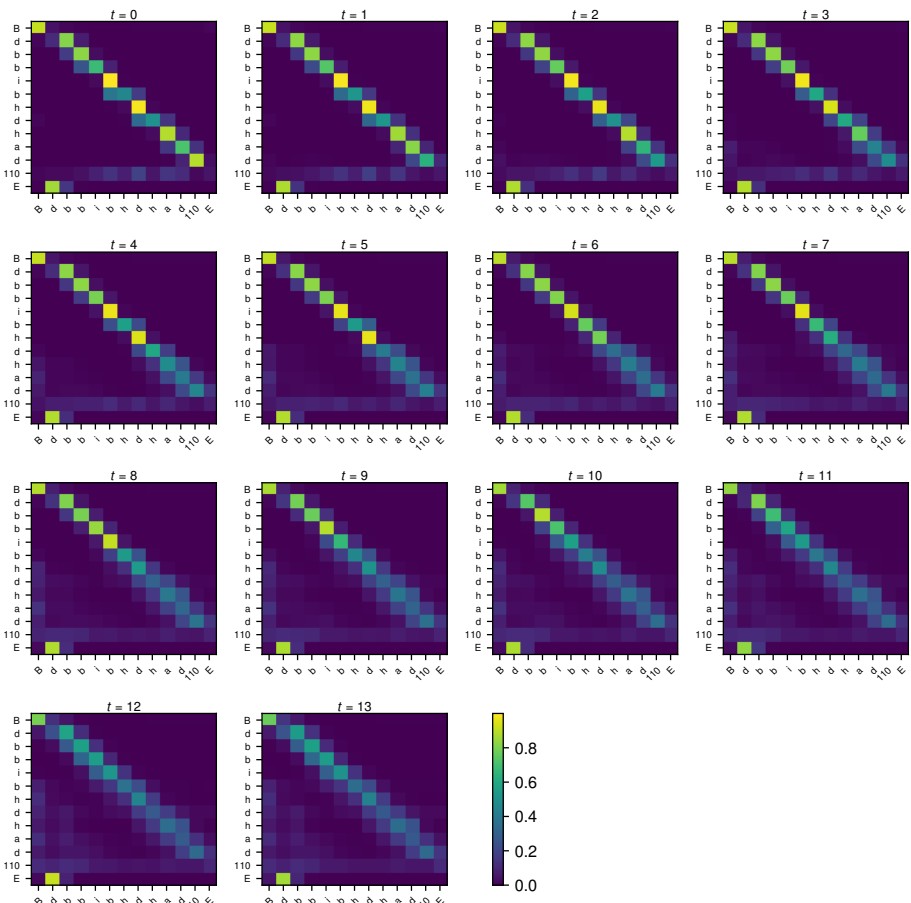

Figure 8: Attention map for every computational step for a Transformer with gating and relative/absolute positional encoding (presented in Figure 2) on CTL. The attention pattern is relatively stable over time, and it gets blurrier only after the given column is processed and updated. The gate sequence for the same input can be seen in Figure 9.

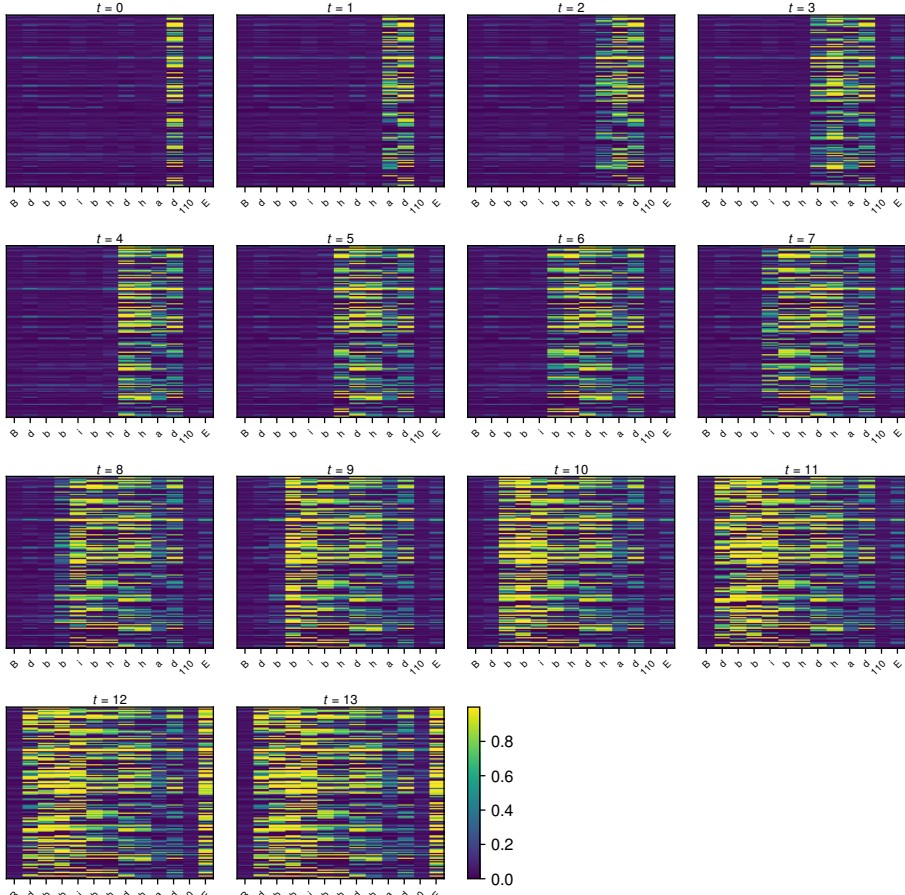

Figure 9: Gates for every computational step for a Transformer with gating and relative/absolute positional encoding on CTL. The gates are closed until all arguments of the given operation become available. The attention maps for the same input can be seen in Figure 8.

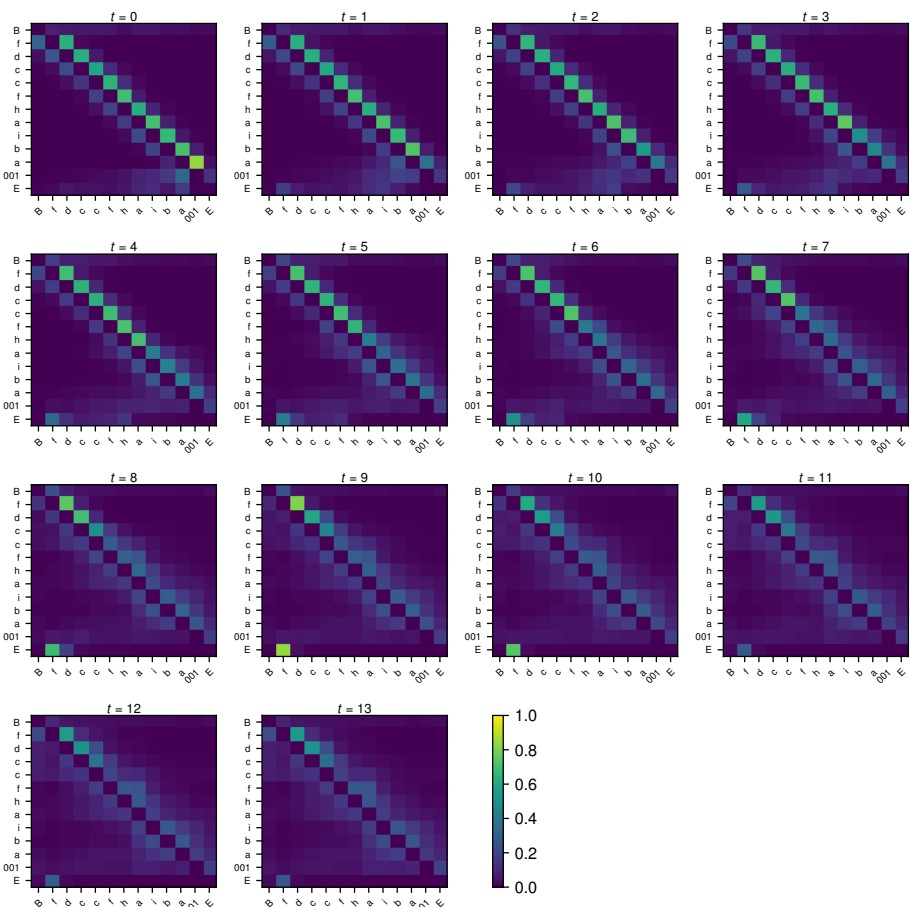

Figure 10: Attention map for every computational step of the NDR on CTL. The network correctly and clearly focuses on the last element of the sequence, and the last sharp read happens in step 10 - corresponding to the 10 function calls in the example. The gate sequence for the same input can be seen in Figure 11.

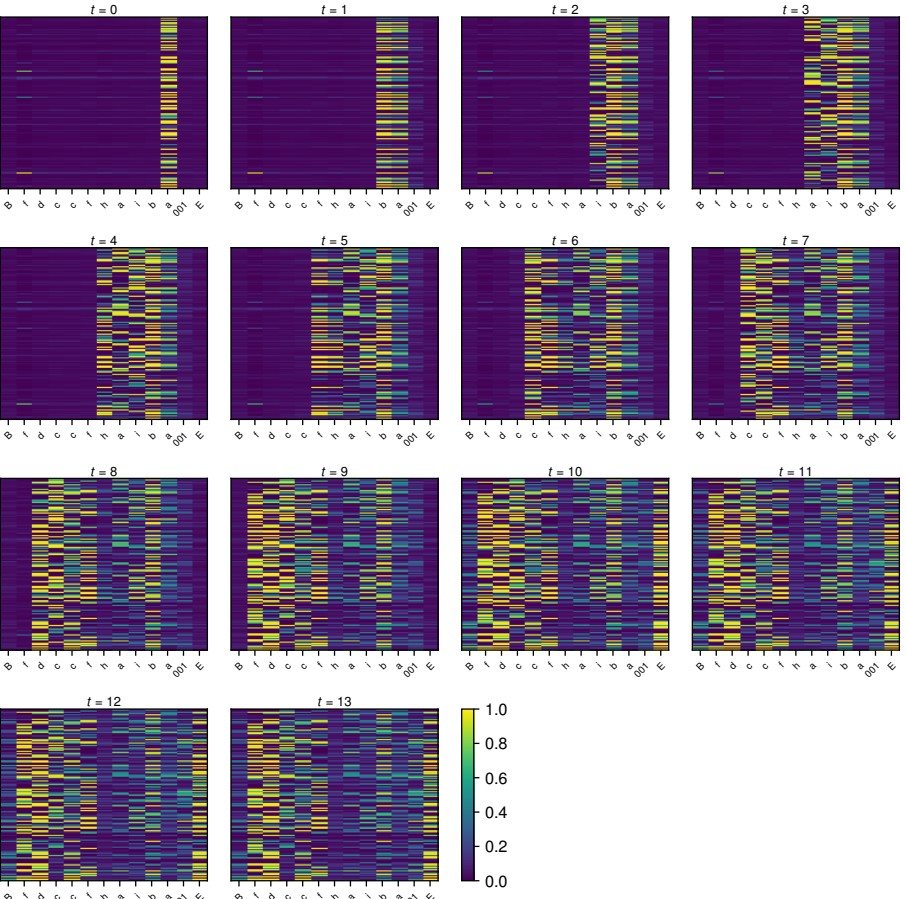

Figure 11: Gates for every computational step of the NDR on CTL. The gates remain closed until all arguments of the given operations become available. The attention maps for the same input can be seen in Figure 10.

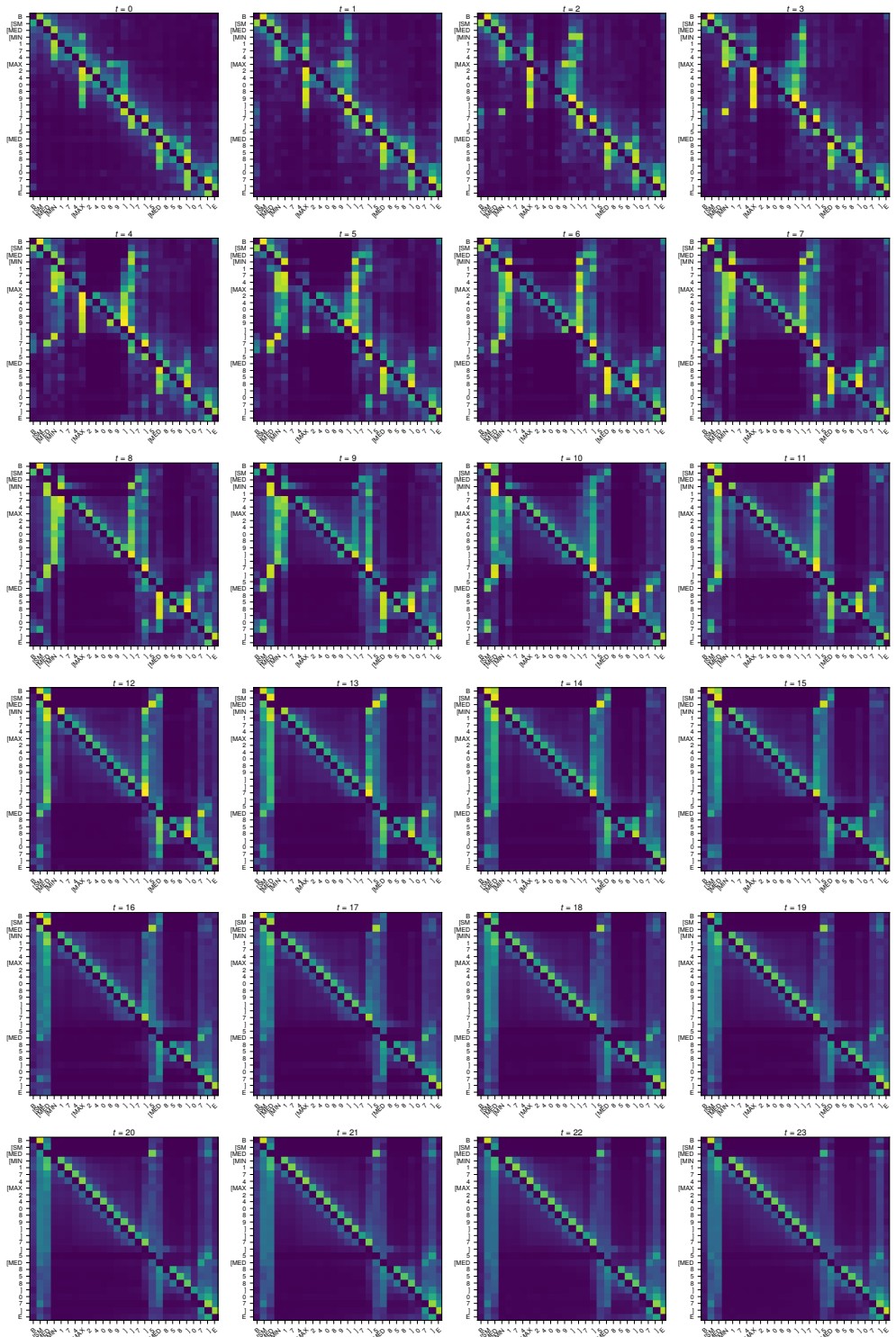

Figure 12: Attention maps for every computational step of the NDR on ListOps. The network has 16 heads; the max of them is shown. The input has only depth 4, which explains the early stopping of the computation, roughly after 8-9 steps, after which the attention barely changes. The corresponding gate maps for the same input can be seen in Figure 14.

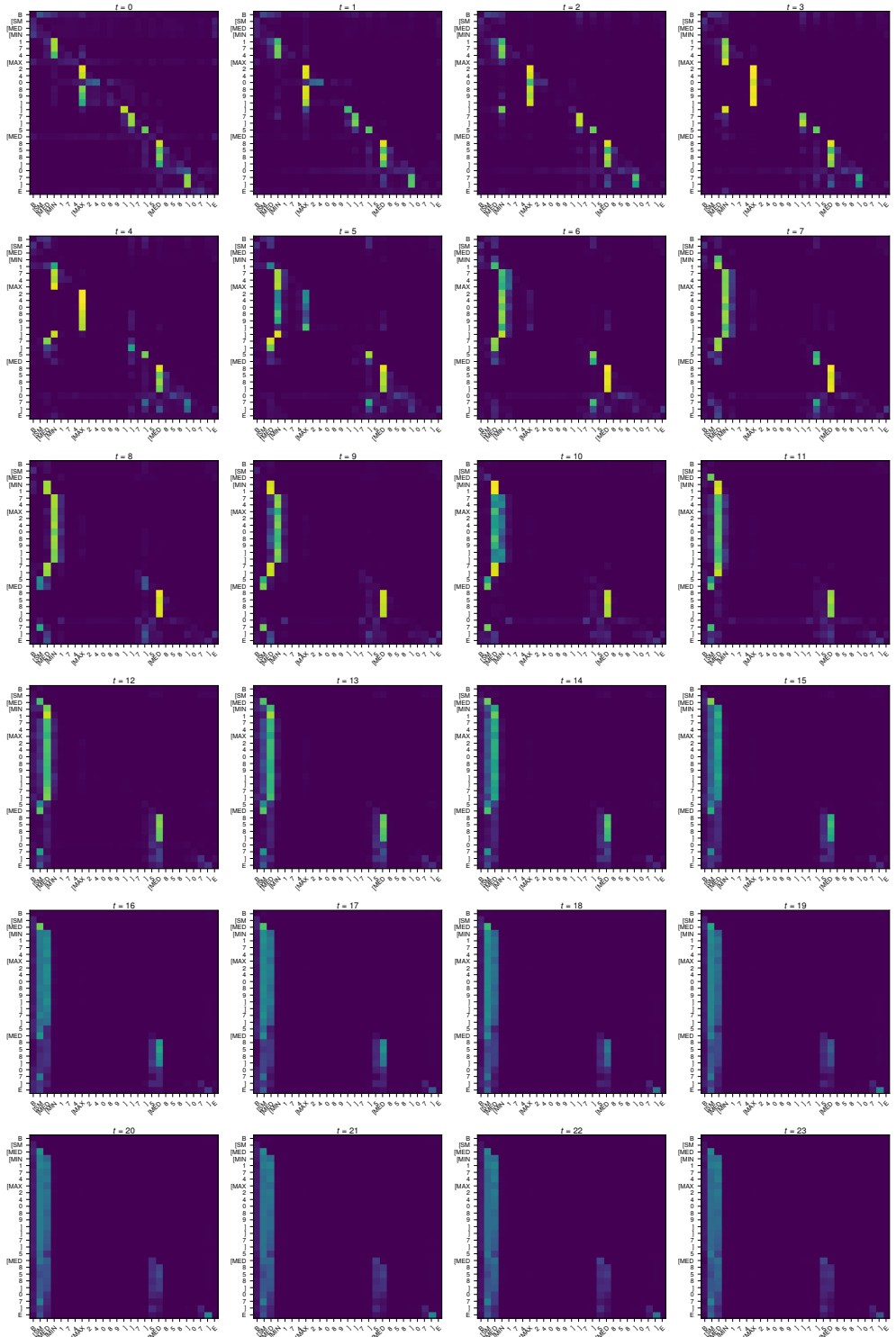

Figure 13: Attention maps for *head 13* of the NDR in every computational step on ListOps. This head shows the operands for each operation. Following it, we observe the hierarchy and the order in which the operations are performed.

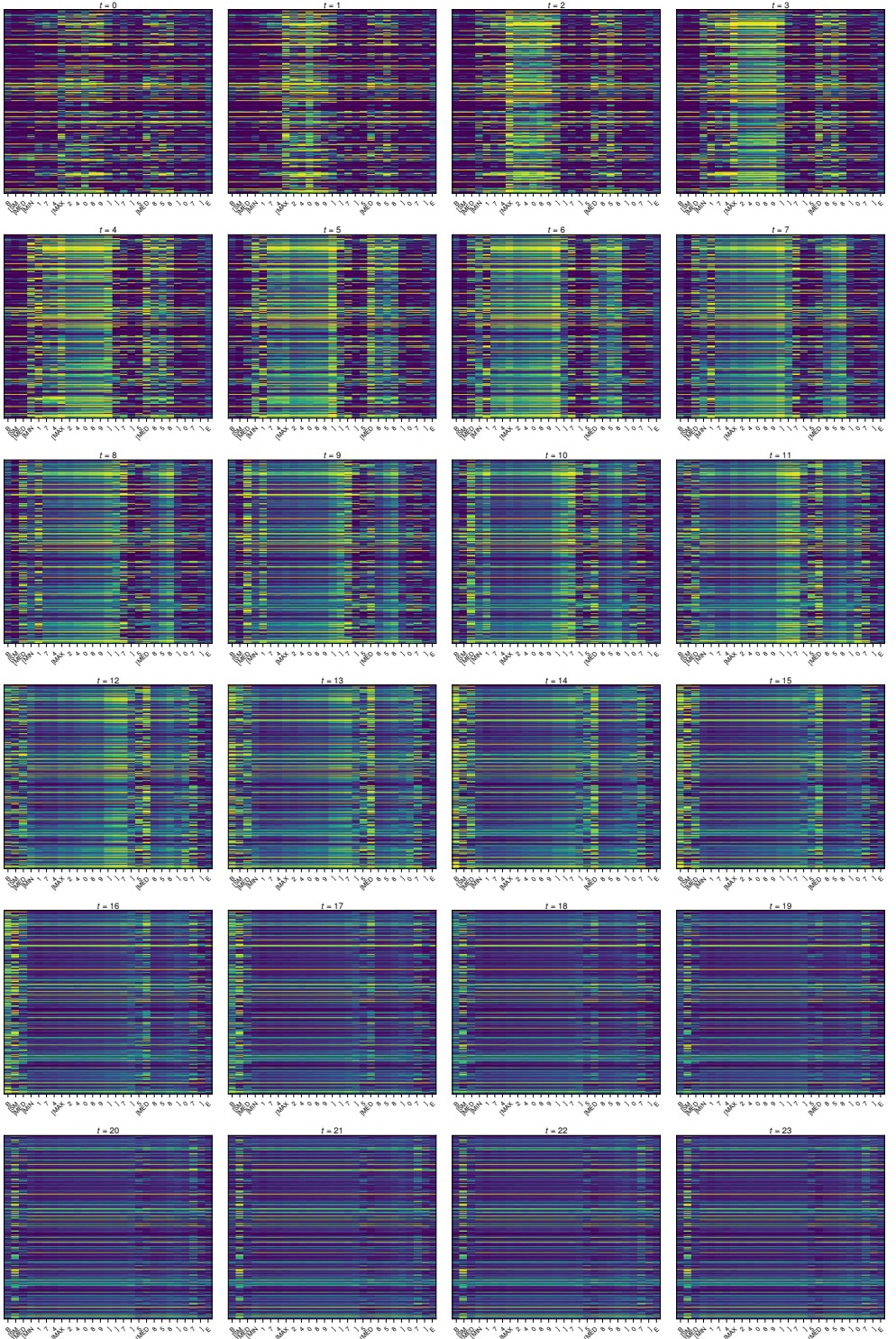

Figure 14: Gates for every computational step of the NDR on ListOps. Gates open for the deepest operations in the tree, processing proceeds upwards in the computational tree. The input has only depth 4, which explains the early stopping of the computation, roughly after 8-9 steps. The attention maps for the same input can be seen in Figure 12.

