# OpenReview forum: "The Neural Data Router: Adaptive Control Flow in Transformers Improves Systematic Generalization"
_ICLR.cc/2022/Conference — ICLR 2022 Poster_

### Official Review · Reviewer_zmLb · 2021-10-26

**Correctness:** 3
**Technical Novelty And Significance:** 3
**Empirical Novelty And Significance:** 2
**Recommendation:** 8
**Confidence:** 4

**Main Review:**


The paper is very clearly written, and proposes an interesting solution to an important question. The tasks chosen are meaningful, and the experimental results suggest that the proposed architecture can solve the extrapolation problem. The technical aspects are precisely documented, which makes the research easy to reproduce.

My main concerns are related to the experimental comparisons, and the impact of certain design choices, such as the use of a fixed model depth at training and test time, the use of the last word in the encoder representation as the basis for model output, and the absence of the Adaptive Computation Time (ACT) in the Universal Transformer implementation that serves as the main baseline. This makes it difficult to judge the impact of the two improvements suggested (copy-gate and geometric attention), and the benefits of the new architecture, compared to an encoder-only state-of-the-art version of the Universal Transformer (with relative positional embedding, and ACT). I believe improving this part of the experimental design and discussion would greatly reinforce the paper. Below are my concerns and questions, split into four themes.

*Computational, and model, depth*

At the beginning of section two, the authors argue that four properties are needed for network to extrapolate to larger problems:
- shared layers
- depth of the computational graph
- step skipping
- short-range attention

I would disagree with the second point, for two reasons. First, computational depth is a relative notion. In an arithmetic task, I can choose to represent modular addition as one operation, or two (addition and modulo), or even three (digit addition, carry propagation, modulo). On the other hand, some linear algebra packages define "add and mul" as a single operation. There is no doubt that network depth should somehow increase with complexity, but defining it from computational depth seems unpractical.

Second, since you use shared layers, model depth can be varied without having to retrain. Specifically, model depth could be adjusted to the complexity of the training examples, and then increased at inference to fit the complexity of the test set. Using the maximum depth in both train and test sets (provided it can be defined from computational trees) is not necessary and might not even be beneficial.

In a recent paper (https://arxiv.org/abs/2106.04537), Schwarzschild et al. have shown (using different architectures, and testing on different tasks) that adding iterations during inference could help models extrapolate to larger problems. It would be interesting to test this on TCF (and baselines).

*Copy-gating and variable depth*

The original Universal Transformers paper proposes a copy-gating mechanism, which uses the Adaptive Computation Time (ACT) mechanism (Graves 2016) at the token level. The gating works differently than in TCF: all gates begin closed, and once opened remain so. However, I believe this (universal transformer plus token-level ACT gating) is the correct baseline for TCF. Can such a comparison be provided? This is all the more important as gating has a large impact on performance, for the three problems considered.

ACT-gating has another merit: it adaptively controls the depth of the Universal Transformer, which goes on iterating until all gates are open. This means that the model can adjust to longer sequences by iterating for more "ponder time". This would provide an adaptive solution to the depth adjustment problem discussed in the previous section. Do you think an adaptive control for the number of iterations/layers could be implemented?

*Encoder-only output, and geometric attention*

The original Universal Transformer is a sequence to sequence model. When decoding a solution, all the output sequence of the encoder is attended to. In your implementation, only the last element in the output is taken into account. As you observe in the results discussion of section 3.1, this creates problems at test time because the position of the last word changes as sequence length increases. It also complicates training, since the output position depends on the sequence length (which varies in all the problems you propose). As you show, this can be alleviated by relative positional embeddings and directional encodings, which can be used to force the result to "move right" as the computation proceeds. It also seems to be the main justification of geometric attention (which seems to bring very little when used alone, cf table 1).

But could the original problem, variable output position, be eliminated? What would happen if the output position had a fixed positional embedding? This could be done in many ways: reading the output from the first position instead of the last (since the transformer is bidirectional, this should have no adverse effect), or from some other fixed value (e.g. the fifth output word), or enumerating positions so that the last token has a fixed embedding (e.g. counting backward, or from both ends to the center).

Another question is the use of a single output element to decode the solution. Would the model be improved by using a longer part of the output sequence (e.g. the N first output words, with N the minimal size of output sequences, or shorter output padded to this size)?

An alternative (and in my opinion much better) solution would be to use a special attention-based layer for the decoder (an attention plus a linear layer working from the output sequence of the encoder). This amounts to a minimal seq2seq model, with one non-shared, cross-attention-only layer in the decoder. This would eliminate the variable output position problem, and allow the full output sequence to be taken into account while decoding.

I believe these alternative decoders need to be tested. Without them, it is hard to assess the importance of relative positions and geometric  attention.

*Compositional table lookup: the backward case*

Unless their architectures are bidirectional, the backward task is very unfair on LSTM and DNC, which are causal models. To solve the backward task, they would need to memorize all the tables before seeing the value to be operated on, an impossible task given their capacity.

Transformers, on the other hand, are bidirectional. Their bad performance on the IID backward case comes as a surprise, and the unusually large error on this observation suggests an experimental problem. Do you have explanations about this high standard deviation of experimental results?

On the test data, the fact that relative positional embeddings seem to improve the forward, but not the backward case, might be due to the choice of the last term in the output as the result to be decoded. Would, for instance, the results be inverted if the first output were chosen instead (or some fixed middle position)?

Overall, I am not certain this backward case helps the argument about the efficiency of TCF (what it demonstrates, I think, is that causal models like the LSTM need their inputs to be presented in a particular order, which is no new news...).


**Summary Of The Paper:**


The authors propose Transformer Control Flow (TCF), a set of improvements to the Universal Transformer (Dehghani et al, ICLR 2019). They show that, for three compositional problems, TCF allows trained models to generalize to longer sequences, a common problem of many transformer implementations.

As in the Universal Transformer (UT), the encoder consists of one shared transformer layer (self attention + fully connected network) which is iterated through a fixed number of times, by feeding the output of each iteration back into the input of the shared layer. However, whereas the UT uses a sequence to sequence model, TCF is an encoder-only architecture, which decodes the last element in the output sequence as the final result.

Two new features are introduced :
- a gating mechanism that allows the model to "skip a layer" (the input is then copied to the output), on the basis of the self-attention output,
- a weighting system for the outputs of attention heads, which favors short-range attention (i.e. tokens close to the one currently considered), and can be trained to be biased towards a certain direction (before or after the current token).

Experiments are conducted over three tasks:
- predicting the output of sequences of permutations of 8 elements, in prefix or postfix notation,
- predicting the result of of additions and multiplications modulo 10, in infix notation,
- predicting the result of operations on lists of small integers, in prefix notation.

For each task, TCF is shown to be capable of extrapolation to larger problems (i.e. longer sequences) than those seen at training.

**Summary Of The Review:**


TCF appears to be a promising architecture, and the experimental results seem good. However, some design and experimental choices make them difficult to compare with the original Universal Transformers. In particular, the choice of a fixed depth at train and test time might be an unnecessary constraint, the proposed gated-copy operation should be compared with the ACT-based copy described in the UT paper, and the use of the last element in the output  sequence as the basis for decoding might unduly increase the necessity of relative positions and short-range attention (ie the geometric attention and directional encoding recommended by the authors).

Hence my note of 6, which I would gladly increase if additional experimental results are provided.

Edit: Thank you very much for the detailed response. I have raised my note to 8.

---

> ### Author Response · Authors · 2021-11-12
> **Response to Reviewer zmLb, part 1/3**
>
> We thank the reviewer for valuable feedback.
> Please find our response inline:
>
> > *Computational, and model, depth*
>
> > *At the beginning of section two, the authors argue that four properties are needed for network to extrapolate to larger problems:*
> > *shared layers, depth of the computational graph, step skipping, short-range attention.*
>
> > *I would disagree with the second point, for two reasons. First, computational depth is a relative notion. In an arithmetic task, I can choose to represent modular addition as one operation, or two (addition and modulo), or even three (digit addition, carry propagation, modulo). On the other hand, some linear algebra packages define "add and mul" as a single operation.*
>
> We first would like to clarify that merging multiple operations into one is exactly what we wish to avoid. We do not want the model to learn one merged operation "add mul" but separate "add" and "mul" operations instead, such that they can be composed to generalize to both "add and mul" and "mul and add" orders. That having said, if the reviewer finds the word "computational graph" confusing, we could replace it by "computation".
>
> > *Second, since you use shared layers, model depth can be varied without having to retrain. Specifically, model depth could be adjusted to the complexity of the training examples, and then increased at inference to fit the complexity of the test set. Using the maximum depth in both train and test sets (provided it can be defined from computational trees) is not necessary and might not even be beneficial. In a recent paper (https://arxiv.org/abs/2106.04537), Schwarzschild et al. have shown (using different architectures, and testing on different tasks) that adding iterations during inference could help models extrapolate to larger problems. It would be interesting to test this on TCF (and baselines).*
>
> In the ListOps experiment with our TCF model, we actually already did exactly what the reviewer suggests.
> The model is trained with a number of layers which is determined from a heuristic based on the computational depth from the training set.
> At test time, the number of computational steps is increased.
> We observed that doing this does not affect the model performance compared to the case where we train the model with the depth necessary for the test set. This allowed us to speed up training.
> The corresponding descriptions were effectively missing in the main text.
> We will add them in the updated text.
> We thank the reviewer for drawing our attention to this.
>
> > *Copy-gating and variable depth*
>
> > *The original Universal Transformers paper proposes a copy-gating mechanism, which uses the Adaptive Computation Time (ACT) mechanism (Graves 2016) at the token level. The gating works differently than in TCF: all gates begin closed, and once opened remain so. However, I believe this (universal transformer plus token-level ACT gating) is the correct baseline for TCF. Can such a comparison be provided? This is all the more important as gating has a large impact on performance, for the three problems considered.*
>
> We first would like to clarify that the "copying" mechanism resulting from the ACT (stop computation from a certain time = copy for the remaining time) is fundamentally different from our copy gate. Our copy gate allows Transformer columns to keep the input unchanged until it’s their turn to be processed (a crucial property to implement control flow like behavior). This behavior can not be simulated by the ACT.
>
> > *ACT-gating has another merit: it adaptively controls the depth of the Universal Transformer, which goes on iterating until all gates are open. This means that the model can adjust to longer sequences by iterating for more "ponder time". This would provide an adaptive solution to the depth adjustment problem discussed in the previous section. Do you think an adaptive control for the number of iterations/layers could be implemented?*
>
> We fully agree with the reviewer that augmenting our model with the ACT would be interesting. A more recently proposed version (PonderNet; Banino et al. 2021, which we already cite) might be a better candidate, since the original version is well known to be difficult to tune/stabilize.
> However, implementing and tuning it would require an entirely new set of experiments which would require some time (infeasible within the rebuttal period).
> While we can argue that this is orthogonal/complementary to the methods we propose, if the reviewer finds this
> to be crucial, we would be happy to run and include
> the corresponding experiments for the camera-ready version (in case the paper gets accepted).

---

> > ### Author Response · Authors · 2021-11-12
> > **Response to Reviewer zmLb, part 2/3**
> >
> > > *Encoder-only output, and geometric attention
> >
> > > *The original Universal Transformer is a sequence to sequence model. When decoding a solution, all the output sequence of the encoder is attended to. In your implementation, only the last element in the output is taken into account. As you observe in the results discussion of section 3.1, this creates problems at test time because the position of the last word changes as sequence length increases. It also complicates training, since the output position depends on the sequence length (which varies in all the problems you propose).
> > As you show, this can be alleviated by relative positional embeddings and directional encodings, which can be used to force the result to "move right" as the computation proceeds. It also seems to be the main justification of geometric attention (which seems to bring very little when used alone, cf table 1).
> > But could the original problem, variable output position, be eliminated?
> > What would happen if the output position had a fixed positional embedding? This could be done in many ways: reading the output from the first position instead of the last (since the transformer is bidirectional, this should have no adverse effect), or from some other fixed value (e.g. the fifth output word), or enumerating positions so that the last token has a fixed embedding (e.g. counting backward, or from both ends to the center).*
> >
> > We will provide an additional experimental result
> > for the variant where the readout is on the first position.
> >
> > > *Another question is the use of a single output element to decode the solution. Would the model be improved by using a longer part of the output sequence (e.g. the N first output words, with N the minimal size of output sequences, or shorter output padded to this size)? An alternative (and in my opinion much better) solution would be to use a special attention-based layer for the decoder (an attention plus a linear layer working from the output sequence of the encoder). This amounts to a minimal seq2seq model, with one non-shared, cross-attention-only layer in the decoder. This would eliminate the variable output position problem, and allow the full output sequence to be taken into account while decoding.
> > I believe these alternative decoders need to be tested. Without them, it is hard to assess the importance of relative positions and geometric attention.*
> >
> > We'd like to clarify that all tasks considered here require a single output prediction. We thus should be able to work with a single fixed readout column (which has access to the entire context via self-attention).
> > We actually did try the variant proposed by the reviewer
> > with an additional cross-attention layer for the readout.
> > However, the generalization performance was not better, which is not surprising since such a layer does not fundamentally address the problem of length generalization.
> >
> > > *Compositional table lookup: the backward case*
> >
> > > *Unless their architectures are bidirectional, the backward task is very unfair on LSTM and DNC, which are causal models. To solve the backward task, they would need to memorize all the tables before seeing the value to be operated on, an impossible task given their capacity.*
> >
> > The main purpose of Table 1 is to compare Transformer variants.
> > By reporting the performance of uni-directional LSTM/DNC, our sole goal was to illustrate that the presentation order matters.
> > A left-to-right LSTM can perfectly solve the forward version of the task, just because its inductive bias matches this specific problem.
> > For an arbitrary task, the order of processing is not given.
> > For example, for ListOps, the processing should start from the deepest point in the parse tree, which is probably somewhere in the middle of the sequence.
> > Our goal is not to tailor a model with an inductive bias for a problem specific processing ordering for each task.
> > An ideal model should be able to solve the task independently of the presentation order.
> > For the CTL task, that means that the same model should solve the problem (at least) in forward and backward orderings.
> > To better illustrate this, we will add results for the bidirectional LSTM to the updated version, which should be able to solve the CTL task in both directions but not the other tasks.
> > We'd like to stress that our goal is not to engineer solutions which solve specific tasks.
> > We are using these datasets to evaluate
> > potential candidates for the most generic solution.

---

> > > ### Author Response · Authors · 2021-11-12
> > > **Response to Reviewer zmLb, part 3/3**
> > >
> > > > *Transformers, on the other hand, are bidirectional. Their bad performance on the IID backward case comes as a surprise, and the unusually large error on this observation suggests an experimental problem. Do you have explanations about this high standard deviation of experimental results?*
> > >
> > > In practice (see, e.g., Dubois et al. 2020, whom we cite), the performance of neural networks across seeds can highly vary on some algorithmic tasks,
> > > when there is some sub-optimal choice in the architecture (the performance can jump when the optimization finds the "right" solution, and models with a sub-optimal architecture often fail to find the right solution consistently across seeds).
> > > We have an illustrative example here: simply introducing the relative positional encoding fixes the issue: our "Transformer + rel" baseline is stable in both directions.
> > >
> > > > *On the test data, the fact that relative positional embeddings seem to improve the forward case, but not the backward case, might be due to the choice of the last term in the output as the result to be decoded. Would, for instance, the results be inverted if the first output were chosen instead (or some fixed middle position)? Overall, I am not certain this backward case helps the argument about the efficiency of TCF (what it demonstrates, I think, is that causal models like the LSTM need their inputs to be presented in a particular order, which is no new news...).*
> > >
> > > We do not argue the efficiency of TCF only based on the backward case.
> > > As we already mentioned above, we claim that a good model should succeed at this task independently of the presentation order.
> > > For example, in Table 1/Longer, "Transformer + rel + gate" performs well on the forward presentation while failing on the backward one. In contrast, TCF works well in both directions
> > > of the CTL, as well as on other tasks.
> > > We'd like to stress once again that our goal is to achieve a generic solution which can solve arbitrary problems with an arbitrary underlying input processing order.
> > >
> > > We hope our response clarifies the spirit of this work and successfully resolved many of the reviewer’s concerns.
> > > If such is the case, we'd greatly appreciate it if the reviewer would consider increasing the score. Thank you.

---

> ### Author Response · Authors · 2021-11-19
> **Thank you**
>
> Thank you very much for the updated score.

---

### Official Review · Reviewer_TUV2 · 2021-10-31

**Correctness:** 4
**Technical Novelty And Significance:** 4
**Empirical Novelty And Significance:** 4
**Recommendation:** 8
**Confidence:** 4

**Main Review:**

Overall, I enjoyed reading this work. The writing is clear and to the point, and the approach itself is very well motivated (see questions for more), and simple to implement without too many tunable hyperparameters. And as such, the experiments are cleanly setup and do suggest improved improved generalization. From the analysis of the attention maps, we can see that the method is doing exactly what it is supposed to as well (that is, copying previous values until other intermediates have been computed, paying more attention to local hidden states etc). Based on these strengths, I recommend that this paper be accepted to the conference.

So now, let me focus on some weaknesses / suggestions / questions:

Overall positioning: Firstly, I think the paper should probably make it more clear that it’s only focusing on a very specific notion of systematicity that has to do with length / depth generalization, and not other more traditional notions like generalizing to new compositions (which isn’t really something that is evaluated) like SQOOP from Bahdanau 2019 etc.

Evaluation: Secondly, while not a strict requirement, there is no evaluation on language tasks / pseudo language tasks like SCAN - there is a length generalization benchmark within SCAN itself and it would be good to know how this method does on that.

Analysis: In Figure-2 bottom, what does It is unclear what the y axis is. Isn’t the copy gate just a single number for each time step, for each layer? If so, i would’ve expected the figure to just be a single number for each time step for the various layers, so I don’t understand what the grid signifies.


**Summary Of The Paper:**

This paper proposes two modifications to provide additional inductive bias to the attention mechanism in the transformer architecture. The first modification adds a copy mechanism to simulate a “no-op” at a given transformer layer, and the second modification is an attention mechanism that is biased towards attending to local context. Both of these modifications are motivated as being useful for algorithmic tasks like compositional table lookup and arithmetic. From experiments that are *mostly* concerned with some kind of length/depth generalization, we see very significant improvements.


**Summary Of The Review:**

Overall, I think based on the results I recommend that the paper be accepted into the main conference. The problem is well motivated, the comparisons are fair and the results compelling though there is some scope for improvement that i highlight in the weakness section of the main review

---

> ### Author Response · Authors · 2021-11-12
> **Response to Reviewer TUV2**
>
> We thank the reviewer for many positive comments and valuable feedback.
>
> > *Overall positioning: Firstly, I think the paper should probably make it more clear that it’s only focusing on a very specific notion of systematicity that has to do with length / depth generalization, and not other more traditional notions like generalizing to new compositions (which isn’t really something that is evaluated) like SQOOP from Bahdanau 2019 etc.*
>
> Thank you for pointing this out. We received a similar comment from Reviewer rpDa. In the updated text, we will clarify that we only focus on a specific type of systematic generalization (so-called productivity i.e. length generalization) with a special focus on algorithmic tasks.
>
> > *Evaluation: Secondly, while not a strict requirement, there is no evaluation on language tasks / pseudo language tasks like SCAN - there is a length generalization benchmark within SCAN itself and it would be good to know how this method does on that.*
>
> The reviewer is right to point this out.
> However, we do/will not have such experiments yet.
> In fact, tasks such as SCAN would require a full sequence-to-sequence model, while our current model is designed as an encoder-only model (which is the focus of this work: drawing parallels between adaptive control flow and Transformer encoder columns). We are currently working on the corresponding extensions for the future work.
>
> > *Analysis: In Figure-2 bottom, what does It is unclear what the y axis is. Isn’t the copy gate just a single number for each time step, for each layer? If so, i would’ve expected the figure to just be a single number for each time step for the various layers, so I don’t understand what the grid signifies.*
>
> Here the gate is a vector and applied element-wise. We also tried a version with scalar gating, but we found it harder to train in general.
> We speculate that the vector version is better
> because it allows the model to flexibly put some elements in the state vector in the copy-mode (e.g., some results of a computation to be directly copied to the next layer) while still transforming other components (e.g., some signals to communicate with other columns), assuming that these different types of information are stored in the Transformer state vector.

---

### Official Review · Reviewer_rpDa · 2021-11-03

**Correctness:** 3
**Technical Novelty And Significance:** 3
**Empirical Novelty And Significance:** 3
**Recommendation:** 8
**Confidence:** 4

**Main Review:**

## Main strengths:
1) The paper is well written and does a great job in introducing the problem and revealing the flaws of the universal transformer in achieving good performance in the described tasks, as well the authors' intuition of the properties of a good solution.
2) The main components of the proposed method are explained in sufficient details to help reproducing the proposed method.
3) The proposed benchmarks and datasets, the empirical approach, and chosen hyperparameters are provided and discussed in details.
4) The paper is well positioned with regards to the related work.

## Main Weaknesses:
5) It is not clear if the the considered benchmarks cover all required aspects of task generalization, or the generalization is only valid for tasks that are to some extent similar to the considered experiments. The authors should further explain which aspects, if any, are missing and are not addressed in this work.
6) It is not clear if the considered assumptions are always necessarily and correct. The authors should address the following questions in the paper either in form of justified explanations or if required with ablation studies:

	6.1) Is there any task which would benefit or require settings that are not covered by the considered settings described in section 2?

	6.2) Regarding point 2 in section 2: What if the data dependency graph was too long that memory complexity would not practically allow to use such a depth? In other words, to what extent the proposed depth is necessary?

	6.3) Regarding point 3 in section 2: Could the gating function result in a shortcut/collapse in optimization? (Considering a far more complex task that is generally addressed by transformers could reveal such issues.)

	6.4) Regarding the final point in section 2: Could a task would prefer non-local operations to local ones? Does the performance of the proposed method degrade in that situation?

7) Some previous works, for example on the ListOps task, consider sequences that are orders of magnitude longer than the ones considered in this paper (A couple of examples are [1], [2]). It is not clear if the claim that previous results did not achieve the perfect accuracy is well-supported? It seems like that to be fair, the authors should have considered some of the SOTA methods and adapt their hyperparameters for these tasks with limited sequence length before testing how they would perform.

### Minor points:
8) In section B.2, the set of values or ranges over which the hyperparameters are searched should also be mentioned.
9) First line of page 14. "sample" -> "sampled"

### References:

[1] "Modeling Hierarchical Structures with Continuous Recursive Neural Networks" by Chowdhury, J.R. and Caragea, C., (arXiv:2106.06038v1)

[2] "Nystr ̈omformer: A Nystr  ̈om-based Algorithm for Approximating Self-Attention" by Xiong et al., (arXiv:2102.03902v3)

**Summary Of The Paper:**

This paper addresses an issue of transformers that sometimes they fail to find solutions that are easily expressible by attention patterns. The issue is justified to be the same as the problem of learning useful control flow. The authors propose two modifications, namely adding a copy gate functionality and a geometric attention module which facilitates focusing on local useful operations. The resulting method achieves near perfect accuracy on the considered benchmarks for length generalization, simple arithmetic tasks, and computational depth generalization.

**Summary Of The Review:**

The paper is well organized and covers the background knowledge required to follow the discussions. It motivates the goal, provides justifications for the choices made in proposed methods, follows a thorough empirical approach, and achieves near perfect results in the considered benchmarks. There are a few points of concerns that need to be cleared before I can fully support the submission. Regardless, I see many of the properties of a good research project and so I lean towards accepting the paper at this stage. I look forward to authors' feedback on my concerns before I finalize my decision.

Edit: I thank the authors for responding to my comments in details. After reviewing their response and the changes made in the paper, many of my concerns are resolved. Therefore, I change my recommendation to accept this paper.

---

> ### Author Response · Authors · 2021-11-12
> **Response to Reviewer rpDa, part 1/2**
>
> We thank the reviewer for valuable feedback.
> Please find our response inline:
>
> > *5. It is not clear if the the considered benchmarks cover all required aspects of task generalization, or the generalization is only valid for tasks that are to some extent similar to the considered experiments. The authors should further explain which aspects, if any, are missing and are not addressed in this work.*
>
> The reviewer is right. Here we focus on a specific type of systematic generalization (so-called productivity i.e. length generalization) with a special focus on algorithmic tasks.
> Generalization to novel compositions (systematicity) is for example not studied here.
> This was indeed also pointed out by Reviewer TUV2. We will emphasize this in the updated text.
>
> > *6.1) Is there any task which would benefit or require settings that are not covered by the considered settings described in section 2?*
>
> This is a great question on which we had several rounds of discussion at an earlier stage of the project.
> In the end, we did not manage to find any particular tasks which clearly violate our assumptions (in contrast, we realized that our hypotheses seem suitable for many well-known tasks).
> Thus, in this work, we opted to develop models which satisfy this specific set of properties. If we find any tasks uncovered by these intuitions in the future, we will address the issues and update the model design accordingly. We also clarify this decision/approach in the paragraph "Bottom up approach for improving model architectures" in Sec. 5.
>
> > *6.2) Regarding point 2 in section 2: What if the data dependency graph was too long that memory complexity would not practically allow to use such a depth? In other words, to what extent the proposed depth is necessary?*
>
> Like any other model, ours is limited by the hardware constraints.
> If we hit such a limit, we would need to introduce methods for reducing memory usage (starting from the most basic approaches such as batch size reduction, gradient accumulation etc...).
> Such a limitation is not specific to our approach. Considering regular Transformers and the existing hardware, we also need such adjustments when handling long sequences or training deep/big models.
>
> Answering the second question:
> the reviewer is right to point out that, while we set up our model to be "sufficiently" deep, we had not investigated the "exact" number of layers which is needed for the model to generalize.
> So we conducted an ablation study for the number of layers on the compositional table lookup task:
>
> |num_layers  | IID Fwd    | IID Bwd    | Longer Fwd    | Longer Bwd    |
>
> |12           |1.00+-0.00|1.00+-0.00|1.00+-0.00|0.99+-0.02|
>
> |10        |1.00+-0.00|1.00+-0.00|0.75+-0.04|0.62+-0.05|
>
> |_8         |1.00+-0.00|1.00+-0.00|0.23+-0.02|0.24+-0.03|
>
> |_6         |1.00+-0.00|0.96+-0.03|0.22+-0.05|0.15+-0.01|
>
> |_4         |0.96+-0.04|0.68+-0.11|0.14+-0.01|0.13+-0.01|
>
> We clearly observe that, while the shallow models can also solve the IID split, only the deep models generalize to the longer problems (here the 12-layer model generalizes almost perfectly, but the 10-layer one doesn't).
> This strongly supports our point 2 in section 2.
> We will add this result to the updated version. Thank you.
>
> > *6.3) Regarding point 3 in section 2: Could the gating function result in a shortcut/collapse in optimization? (Considering a far more complex task that is generally addressed by transformers could reveal such issues.)*
>
> Here we do not fully understand the reviewer’s question. Please correct us if our interpretation is wrong.
> In the regular Transformer, there is no obvious solution to skip an entire layer (consisting of a self-attention layer and feedforward block): it would have to somehow zero out the residual/transformation part while compensating for the layer norm... An explicit parameterization as is done in our gate *facilitates* learning such a copying behavior. If this is what is meant by "shortcut/collapse in optimization", then the answer is yes.
> However, we are unsure what we gain from such an interpretation here (please let us know if we misunderstand the question).
> We are also not fully sure which "far more complex task" and "issues" the reviewer is referring to, but learning solutions which generalize in algorithmic tasks is considered as a "difficult" problem for the regular Transformer in general.

---

> > ### Author Response · Authors · 2021-11-12
> > **Response to Reviewer rpDa, part 2/2**
> >
> > > *6.4) Regarding the final point in section 2: Could a task would prefer non-local operations to local ones? Does the performance of the proposed method degrade in that situation?*
> >
> > To be specific, our geometric attention is not necessarily only for "local" operations. It can also learn long range attention in principle. In fact, the only inductive bias we introduce is to attend to the "closest match" which is different from attending to tokens at the closest positions.
> > In addition, since the notion of "matching" is determined based on the learned representations of key and query vectors, the resulting attention should be rather flexible.
> > This is well illustrated with the ListOps example shown in Figure 3. We effectively find long range attention which is necessary to conduct outer operations, while the attention is "local" for the operations at the deepest level of the nested list.
> >
> > > *7. Some previous works, for example on the ListOps task, consider sequences that are orders of magnitude longer than the ones considered in this paper (A couple of examples are [1], [2]). It is not clear if the claim that previous results did not achieve the perfect accuracy is well-supported? It seems like that to be fair, the authors should have considered some of the SOTA methods and adapt their hyperparameters for these tasks with limited sequence length before testing how they would perform.*
> >
> > When we refer to "previous results", we refer to the generic Transformer models (since our goal is to improve them). We will make this clearer in the updated text. Now, the Transformer variant proposed in [2] has a similar performance as the baseline Transformer on ListOps (Table 3 in [2]), and [1] is not a Transformer model. To the best of our knowledge, the regular (and Universal) Transformer is a reasonable Transformer baseline for this task. Regarding the sequence lengths, we agree with the reviewer that they are shorter in our setting, e.g., compared to the Long Range Arena, but they turn out to be sufficient for the purpose of demonstrating the performance gap between the regular Transformers and the proposed methods,
> > which is our main focus.
> >
> > > *Minor points:*
> >
> > > *8. In section B.2, the set of values or ranges over which the hyperparameters are searched should also be mentioned.*
> >
> > We will add the requested numbers in the updated version.
> >
> > > *9. First line of page 14. "sample" -> "sampled"*
> >
> > We will correct this in the updated version.
> >
> > We hope our response provides clear answers to the reviewer's questions. If this is the case, we would really appreciate it a lot if the reviewer could consider increasing the score. Thank you.

---

> ### Author Response · Authors · 2021-11-22
> **Thank you**
>
> Thank you very much for the updated score.

---

### Author Response · Authors · 2021-11-12
**General Response**

We thank all reviewers for valuable time spent on reviewing our work. Please find our response in the individual replies.
We will upload the updated paper next week, once we obtain the results for additional experiments requested by the reviewers.

---

> ### Author Response · Authors · 2021-11-19
> **Manuscript updated**
>
> We uploaded the updated manuscript and supplemental materials containing code used to run requested experiments.
>
> In the updated manuscript, all changes are highlighted in blue.
>
> The reviewers should be able to find all additional experimental results and edits we promised in the individual replies, including
> a new Appendix A containing three ablation studies (on the number of layers, the readout position, and the baselines with ACT which we managed to run in time),
> the bi-directional LSTM baseline for all tasks (Table 1 to 3), and a new Appendix C.1 describing the heuristics on the number of layers used in our experiments.
> Some minor edits were also applied to comply with the page limit.
>
> We thank all the reviewers once again for the time spent reviewing our work.

---

### Public Comment · ~Róbert_Csordás1 · 2022-02-27
**Camera-ready uploaded**

We uploaded the camera-ready version. We note that we changed the model name from "Transformer Control Flow (TCF)" to "Neural Data Router (NDR)", and updated the title accordingly.

---

### Public Comment · ~Róbert_Csordás1 · 2022-05-05
**Camera-ready updated**

Based on the received feedback, we updated our paper for the final version (new figures, fixed typos, improved notations).

---

### Decision · Program_Chairs · 2022-01-20

**Decision:**

Accept (Poster)

**Comment:**

This work proposes a novel Transformer Control Flow model and achieves near-perfect accuracy on length generalization, simple arithmetic tasks, and computational depth generalization.  All reviewers give positive scores. AE agrees that this work is very interesting and has many potentials. It would be exciting if the author could extend this framework to more challenging tasks (e.g. visual reasoning [1. 2]).  Given the novelty of the proposed model, AC recommends accepting this paper!

[1]  CLEVR: A Diagnostic Dataset for Compositional Language and Elementary Visual Reasoning. ICCV 2017.

[2] PTR: A Benchmark for Part-based Conceptual, Relational, and Physical Reasoning, NeurIPS 21